# Vulnerability in research ethics: A systematic review of policy guidelines and documents

Asia Grigis[1,2]☯, Giorgia Beretta[3]☯, Pascal Borry [ID][4], Virginia Sanchini [ID][3,4]*

**1** University of Milan, Milan, Italy, **2** European Institute of Oncology IRCCS (IEO), Milan, Italy, **3** Department of Oncology and Hemato-Oncology, University of Milan, Milan, Italy, **4** Department of Public Health and Primary Care, Centre for Biomedical Ethics and Law, KU Leuven, Leuven, Belgium

☯ These authors contributed equally to this work.
\* virginia.sanchini@unimi.it

## Abstract

### Background

The history of research involving human subjects has demonstrated the importance of offering everyone an equal opportunity to participate in research, while safeguarding those who require special attention. When it comes to vulnerable individuals, this consideration is relevant. However, disagreement still exists about the meaning of vulnerability, the identification and definition of vulnerable populations, and how these concepts should be operationalised in policy documents in order to implement appropriate, preventive and respectful measures for all those subsumed within this category.

### Objectives

Following the PRISMA-Ethics guidance, we performed a systematic review of policy documents to provide a comprehensive overview of how vulnerability is conceptualised and operationalised in research ethics. The aim is to investigate the meaning and definition of vulnerability in research ethics, its normative justification, the comprehensive set of subjects it refers to, and consequent provisions.

### Methods

Our search centred on three main sources: three overview lists that provide comprehensive coverage of research ethics policy documents and guidelines (the *International Compilation of Human Research Standards,* the *Listing of Social-behavioral Research Standards* and the *Ethics Legislation, Regulation and Conventions*); search databases (PubMed and Web of Science) and grey literature (Google Scholar), to ensure completeness of included policy documents. Search strings were developed by the last author (VS) in consultation with the co-first author (GB). The whole screening process was performed by the first (AG) and co-first author (GB),

**Data availability statement:** All relevant data are within the article and its Supporting Information files.

**Funding:** This research is part of the project "Emerging Technologies and Vulnerabilities in Aged Care" (ElderTech), funded by Fondazione Cariplo under the call "Social Research: Science, Technology, and Society" (Grant number: 2020-1322). The funders had no role in the study design, data collection and analysis, decision to publish, or preparation of the manuscript.

**Competing interests:** The authors have declared that no competing interests exist.

separately. The search was originally performed in April 2023, and then re-performed in May 2025 to exclude obsolescent results. English-language policy documents in the field of human research ethics and addressing the subject of vulnerability were included. Eligibility criteria covered both national and international application. For data analysis and synthesis, the authors followed the steps of the QUAGOL methodology: policy documents' reading (AG), highlighting of relevant parts (AG), development of a summary of each document's highlighted parts (AG), summary evaluation and verification against previous QUAGOL steps (AG, GB, VS), creation of a comprehensive scheme (AG, GB, VS), and description of results (AG, GB, VS). No automation tools were used at any stage of the review.

## Results and discussion

Seventy-nine policy documents were included in the review. Research findings were organised in four different subsections, corresponding to four research questions. The analysis of such a significant number and variety of documents allowed us to identify several recurring patterns that characterise the way vulnerability is described and addressed by policy documents. Based on our roles as bioethicists, research ethicists, biotechnologies expert in clinical trials, and study coordinators, we identified some key themes, e.g., a tendency to identify and define vulnerable groups, rather than providing a general definition of vulnerability, and a tendency to define vulnerability in relation to informed consent.

## Conclusions

Only a proper understanding of the meaning of vulnerability, its implications and its normative justifications will make it possible to ensure a fair and ethically legitimate participation in research for all involved subjects. As to the study limitation, only publications written in English, or officially translated in English, were included in the review.

## Introduction

The notion of vulnerability in research ethics was introduced for the first time in *The Belmont Report* [1], in 1979, by the National Commission for the Protection of Human Subjects of Biomedical and Behavioural Research. In this document, "vulnerable people" are defined as those in a "dependent state and with a frequently compromised capacity to free consent" (e.g., racial minorities, economically disadvantaged people, the very sick, the institutionalized). As of the 1980s, policies and guidelines mentioning the concept of vulnerability and its implications for research were published at national and international levels.

Most published documents articulated as their aim to protect participants from abuses in biomedical research and research-related harm or injury. Following the

imperative of protection, many guidelines originally imposed a strict restriction on the enrolment of potentially vulnerable subjects in clinical research. This resulted in the absence of care options for the vulnerable population, however, thus perpetuating injustice [2].

As suggested by more recent trends, the involvement of vulnerable subjects in research should be supported provided that appropriate precautions are taken. A general agreement on how to implement such careful involvement has yet to be reached, however. Balancing protection with adequate participation of vulnerable subjects is a tough challenge, which all stakeholders involved in the academic debate, as well as in clinical research practice, have been trying to address.

Meanwhile, the field of research ethics has seen the emergence of new tendencies that support a shift from a "category" or "group-based notion" of vulnerability, also known as "labelling approach", to an "analytical approach" of vulnerability [3].

According to the former, a participant is considered vulnerable on the basis of his/her belonging to a group of subjects typically considered as such, for instance children, the elderly, pregnant women, prisoners, subjects suffering from physical and/or mental disabilities, etc. Conversely, the analytical approach focuses on defining the conditions as well as the potential sources of vulnerability (which can be both individual and environmental), providing three main accounts of vulnerability. According to consent-based accounts, vulnerability stems primarily from a lack of capacity to provide free and informed consent with respect to participation in research, due to a variety of conditions, such as undue influence and reduced autonomy. Harm-based accounts refer mainly to the assessment of the risks and benefits for research participants, considering vulnerability as a higher probability of incurring harm during research. Finally, justice-based accounts point at unequal conditions and/or opportunities for research subjects as a source of vulnerability.

Although the analytical approach is nowadays considered theoretically preferable, for being more nuanced and potentially more respectful of the different (research) contexts, research ethics committees still tend to prefer the categoric or group-based notion of vulnerability, as a pragmatically simpler solution to protect vulnerable subjects from misconduct in research.

At the same time, there is still profound disagreement regarding the appropriate definition of vulnerability, the comprehensive set of subjects it refers to and its operationalisation in concrete research contexts, namely the practical provisions that can be implemented to ensure adequate participation in research of those considered potentially vulnerable [4].

In spite of the above, we contend that vulnerability may play a useful role as a regulatory category for research ethics, and that the lack of clear, specific guidance on vulnerability – and, more broadly, the ambiguity around vulnerability in research ethics policy documents – may result in stakeholders dealing with vulnerable subjects in a sub-optimal way. First, although research ethics committees, in their documentation, often require a statement about the enrolment of vulnerable populations, to the best of our experience, they do not provide further guidance to researchers on how to address this request. This, on the one hand, may result in research ethics committees using different lists of vulnerable subjects/criteria for vulnerability, and therefore in a differential treatment of vulnerable research subjects, with potentially inequitable outcomes. On the other hand, researchers with poor or no guidance on vulnerability may end up excluding some populational groups traditionally defined as vulnerable, due to the difficulties in managing them appropriately [5]. Accordingly, some research subjects, the most fragile and vulnerable, may potentially be subjected to underrepresentation or disparity in research, the latter a practice that also more recent policy documents have strongly criticized (e.g., 2024 version of the *Declaration of Helsinki*).

So far, to the best of our knowledge, no study has systematically mapped the concept of vulnerability as it is presented in national and international policies and guidelines regulating research on human subjects. Aiming to fill this gap, the systematic review presented here focuses on an analysis of clinical research guidelines and policy documents, in order to gain a comprehensive overview of the way such documents define and operationalise the concept of vulnerability.

This paper is structured as follows. First, in the methodology section, we outline the framework of our systematic review, including the research questions we address, information regarding the literature search process, and the steps we followed for data extraction and synthesis. Next, in the results section we detail how our research questions are

addressed in the documents reviewed, by analysing the definition of vulnerability they provide, the vulnerable subjects they refer to, the reason why they are considered vulnerable and, finally, the provisions suggested to manage vulnerability in research settings. In the discussion section, we present a critical appraisal of our findings, with the aim of providing a general definition of vulnerability that can help stakeholders to identify sources of vulnerability and protect vulnerable research participants in an appropriate manner.

## Materials and methods

To systematically explore the concept of vulnerability in policy documents, we collected a comprehensive sample of guidelines, regulations and legislations referring to the field of human research ethics. We performed a systematic search following the PRISMA-Ethics guidance for systematic reviews [6] and applying them to the context of policy documents.

### Research questions

We formulated the following interrelated research questions:

1. What is the meaning and definition of vulnerability as explicitly reported in research ethics policy documents?

2. What are the groups/populations identified as vulnerable in research ethics policy documents?

3. What are the normative justifications for vulnerability in research ethics policy documents?

4. What are the consequent provisions for vulnerable populations in research ethics policy documents?

### Search strategy

As to the search strategy, we considered three data sources. First, we collected guidelines from three overview lists developed by authoritative organisations: the *International Compilation of Human Research Standards* (2024 edition) [7], the *Listing of Social-behavioral Research Standards* (2018 edition) from the US Department of Health and Human Services [8], and the *Ethics Legislation, Regulation and Conventions* from the European Commission's Horizon 2020 programme [9]. Secondly, we performed a keyword search on two major databases: PubMed and Web of Science. Thirdly, we conducted a grey literature search on Google Scholar.

In addition, we developed search strings by combining keywords from three groups of organizing concepts (Table 1): Group A referred to the topic under investigation (vulnerability), Group B concerned the domain under consideration (human research), and Group C related to the type of documents considered (policies and guidelines).

Each group concept was expressed in specific database/grey literature search terms in a suitable format for the different database/grey literature queries (Table 2 and Table 3).

Search strings were developed by the last author (VS) in consultation with the co-first author (GB).

Furthermore, we conducted database queries on PubMed and Web of Science, on March 1 and April 3, 2023, respectively, using language filters to identify only articles published in English. We examined grey literature on April 17, 2023, again using English as main filter.

**Table 1. Groups of organizing concepts for searching the literature, and associated database search terms.**

| TOPIC | TYPE OF DOCUMENT | DOMAIN/CONTEXT |
|---|---|---|
| vulnerability, fraility, frailness, fragility, vuln-, frail-, frag- | guideline, regulation, legislation, recommendation, policy, code, declaration, normative document, statement | human-subject research, clinical research, clinical trials, research involving humans, research ethics, ethics of research |

**Table 2. Search strings used for searching databases, stratified by organising concepts.**

**PubMed**

| | | | | |
|---|---|---|---|---|
| ((((((((((vulnerability[Title/Abstract]) OR (vulnerab*[Title/Abstract])) OR (fragility[Title/Abstract])) OR (frailty[Title/Abstract])) OR (frail[Title/Abstract])) OR (fragilit*[Title/Abstract])) OR (frailness[Title/Abstract])) OR (frailties[Title/Abstract])) OR (Frailty[MeSH Terms]) | AND | ((((((((((((((guideline[Title/Abstract]) OR (guidelines[Title/Abstract])) OR (recommendation*[Title/Abstract])) OR (code*[Title/Abstract])) OR (legislation[Title/Abstract])) OR (policy[Title/Abstract])) OR (policies[Title/Abstract])) OR (regulation[Title/Abstract])) OR (regulations[Title/Abstract])) OR (declarations[Title/Abstract])) OR (declaration[Title/Abstract])) OR (norm*[Title/Abstract])) OR (statement[Title/Abstract])) OR (statements[Title/Abstract]) | AND | (((((((human-subject research[Title/Abstract] OR (human subject research[Title/Abstract])) OR (clinical research[Title/Abstract])) OR (clinical trials[Title/Abstract])) OR (research involving humans[Title/Abstract])) OR (research involving human subjects[Title/Abstract])) OR (research ethics[Title/Abstract])) OR (ethics of research[Title/Abstract])) OR (ethics[Title/Abstract]) |

**Web of Science**

| | | | | |
|---|---|---|---|---|
| (((((((TS=(vulnerability)) OR TS=(vulnerab*)) OR TS=(fragility)) OR TS=(frailty)) OR TS=(frail)) OR TS=(fragilit*)) OR TS=(frailness)) OR TS=(frailties) | AND | ((((((((((((TI=(guideline)) OR TI=(guidelines)) OR TI=(recommendation*)) OR TI=(code*)) OR TI=(legislation)) OR TI=(policy)) OR TI=(policies)) OR TI=(regulation)) OR TI=(regulations)) OR TI=(declaration)) OR TI=(declarations)) OR TI=(norm*)) OR TI=(statement)) OR TI=(statements) | AND | (((((((((TS=(human-subject research)) OR TS=(human subject research)) OR TS=(clinical research)) OR TS=(clinical trial)) OR TS=(clinical trials)) OR TS=(research involving humans)) OR TS=(research involving human subjects)) OR TS=(research ethics)) OR TS=(ethics of research)) OR TS=(ethics) |

**Table 3. Search string for grey literature (Google Scholar).**

| |
|---|
| clinical research AND research ethics AND (intitle:guidelines OR intitle:recommendation OR intitle:code OR intitle:policy) AND (intitle:vulnerability OR intitle:vulnerable OR intitle:fragility OR intitle:fragile OR intitle:frailty OR intitle:frailness) |

We collected and organised all search results from overview lists, databases and grey literature in a Microsoft Excel database. Subsequently, we pre-screened and screened all the retrieved guidelines, based on a pre-specified set of inclusion criteria.

Upon submission of the manuscript (July 2024), and after the second review round (May 2025), in order to verify that our work did not contain any obsolescence, we checked whether overview lists had been updated, and we re-performed database and grey literature searches. In this manner, we could verify that no new guidelines relevant to our research questions had been published in the meantime. While the paper was under review, five documents were updated (*International Compilation of Human Research Standards*, *the Declaration of Helsinki*, *Guidelines for Good Clinical Practice E6* (and Integrated Addendums E6-(R3), *Implementing Regulation of the Law of Ethics of Research on Leaving Creatures, Medical Products in Human Medicine Act*). Therefore, the Result section was updated on the basis of the novelties introduced by these revisions.

### Inclusion and exclusion criteria

As shown in Figure 1, we performed the pre-screening and screening process according to the statement and flowchart of the Preferred Reporting Items for Systematic Reviews and Meta-Analysis (PRISMA) [10].

Documents were considered eligible according to the following inclusion and exclusion criteria (Table 4):

### Pre-screening

The pre-screening of policies and guidelines involved a first skimming of the documents and excluding the ones which:

1. were not in English;

2. were not available for consultation;

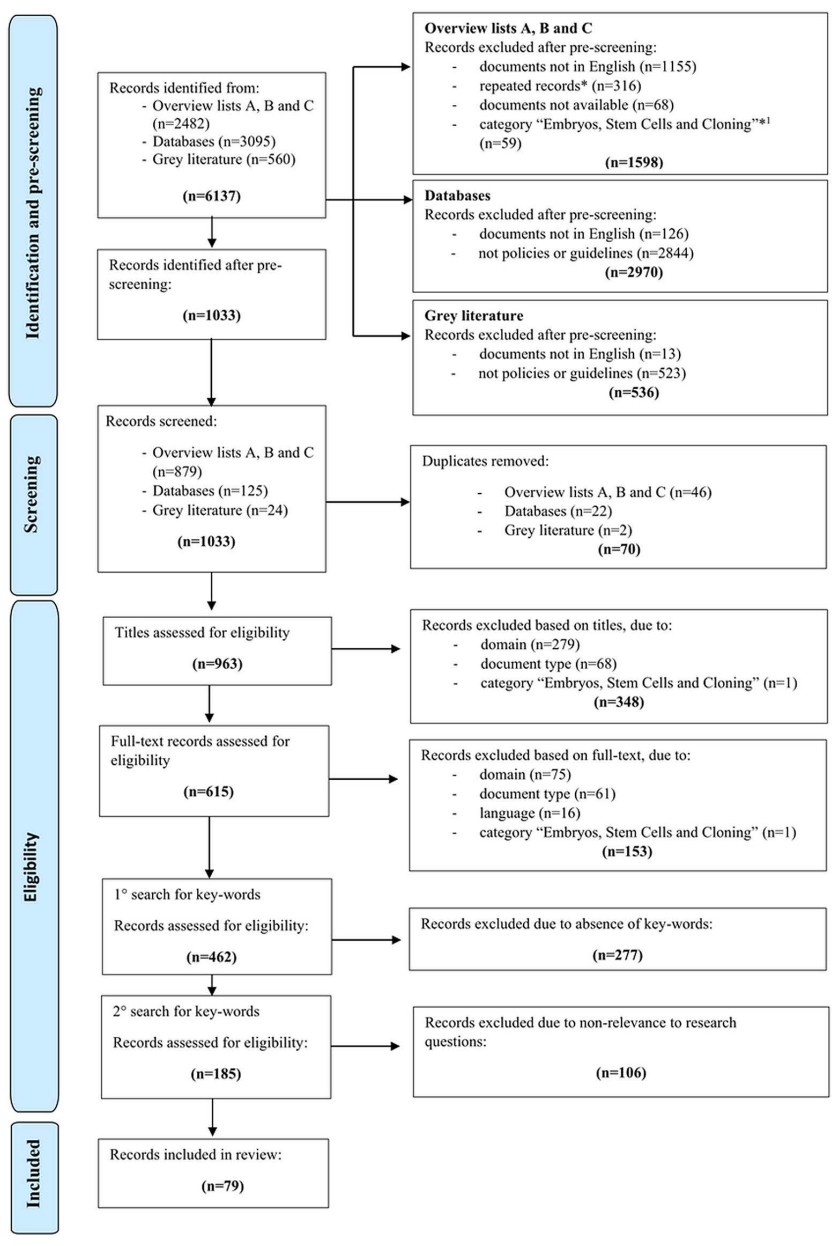

**Fig 1. PRISMA flowchart of pre-screening and screening process.** PRISMA flowchart showing the three data sources search, policies and guide-lines identification, and selection process for the final included documents.

**Table 4. Inclusion and exclusion criteria for selection.**

|  | CRITERIA | |
|---|---|---|
|  | INCLUSION | EXCLUSION |
| Language | Policies and guidelines in English | Policies and guidelines not in English (or not translated into English) |
| Domain | Policies and guidelines related to the field of human research ethics | Policies and guidelines not directly, or indirectly, related to the theme of vulnerability or to the need of specific protection for certain categories of subjects enrolled in the research (i.e., policies regarding medical practice, etc.) |
| Topic | Policies and guidelines addressing the subject of vulnerability in research ethics (i.e., containing the keywords "vulnerability" "fragility", "frailty", "frailness" or words belonging to this word family) | |
| Application | Policies and guidelines with national or international application | |
| Type | Policies and guidelines either legally binding or non-binding | |

3. were reported more than once in the lists, either because they were included in more than one classification made by the authors of the list – General/ Drugs, Biologics and Devices/ Clinical Trial Registries/ Research Injury/ Social-Behavioral Research/ Privacy-Data Protection/ Human Biological Materials/ Genetic Research/ Embryos, Stem Cells, and Cloning – or because, in addition to listing a certain document, different articles or paragraphs of that same document were also listed separately;

4. in *International Compilation of Human Research Standards* (2024 edition) [7] were under the classification "embryos, stem cells and cloning"; these documents were excluded because they fall outside of the scope of our research questions, which pertain to vulnerability when applied to traditional categories of vulnerable subjects, while excluding those whose moral status is being discussed;

5. were retrieved in databases and grey literature, but were clearly not policies and/or guidelines.

## Screening

After the pre-screening process, we screened the remaining documents. First, we left out duplicates manually. Second, the first author (AG) performed a title screening, in line with the inclusion-exclusion criteria reported above, further excluding documents according to the following criteria:

1 Language: despite the language filter applied, some documents not in English were incorrectly included and therefore had to be excluded in the screening phase.

2 Document type: documents which at first appeared to comprise guidelines but were not in fact policy documents were also excluded.

3 Domain: all the documents which did not pertain to the domain of research ethics (i.e., documents regarding clinical practice, diagnosis, treatment etc.) were excluded.

4 Topic: documents which covered only research on embryos and/or stem cells and/or cloning were excluded, because of the reasons reported above.

Thirdly, we screened the text of the documents, to assess the consistency of titles with the actual text and content of the document. By applying the above criteria in this phase as well, a large number of documents were excluded (the complete list of excluded documents may be found in Supporting Information section, S1 Table). Fourthly, we searched the full texts of the remaining documents for these keywords: vulnerability, fragility, frailty and frailness (searched as vuln-, frail-, frag-). The documents not containing any of the mentioned keywords were excluded. Fifthly, we subjected the documents which included the abovementioned keywords to a further process of screening in order to assess whether or not the keywords were used as a recurrent concept in the text and in a context relevant to our four research questions (see above). A complete list of documents excluded from keywords search can be found in the Supporting Information section, S2 Table. We analysed the complete list of included documents by using the QUAGOL methodology (The Qualitative Analysis Guide of Leuven) [11]. For data extraction, the first author (AG) first read and reread several times all the included documents, then highlighting the relevant parts. The same author subsequently developed a summary (QUAGOL scheme) of these highlighted parts. Each scheme was checked separately by the first (AG), co-first (GB), and last (VS) authors to verify that it accurately reported information from each included policy document. This resulted in a comprehensive overview that integrated the most relevant findings, on which basis we synthesised a description of study results (Supporting Information, S3 Table). The full list of QUAGOL schemes can be found in the Supporting Information section, S1 Appendix.

## Results

### General description of the included policy documents

We identified a total of seventy-nine policy documents as eligible to be included in the systematic review. Table 5 presents the complete list of guidelines and policy statements included in this systematic review (identified by Nos. *12–89*, except for *The Belmont Report*, which is numbered [1]), with their key characteristics, including year of publication, issuing organisation and the country where they were adopted.
Direct links to all included guidelines and policy statements can be found in the Supporting Information section, S4 Table.

The identified sample of policy documents comprises nine international documents (accounting for about the 11% of the total) and seventy national documents.

As shown in Table 6, all continents are represented in the list of included guidelines. Europe is the continent with the highest number of issued policy documents (23), even if this number is not significantly higher than that of Africa (19).

As regards the year of publication, the oldest document is *The Belmont Report* [1], published in 1979. Apart from it, only four other guidelines and policy statements were published between 1979 and 2000: three documents appeared in the late 1990s, and one was published in the year 2000. If a higher number of documents was published in the first decade of the 21st century (a total of twenty-six), most documents in fact date from the years between 2011 and 2025.

As to our research questions, policies prove to address Q2, Q3 and Q4 in quite homogeneous ways, while Q1, regarding the meaning and definition of vulnerability as explicitly reported in policy documents, is covered much less (with only ten documents addressing it, accounting for less than 13% of the total).

The most commonly addressed question is Q2: seventy-two documents identify groups/populations as vulnerable in research ethics policy documents.

Furthermore, sixty-eight documents provide some reasons for vulnerability (Q3), which, as reported in our introduction, can be classified into three categories of normative reasons: harm-based, justice-based and consent-based. Some of the documents in our study rely on more than one of these categories: sixty-five documents mention a consent-based reason for vulnerability; twelve documents mention a justice-based reason for vulnerability; thirty documents mention a harm-based reason for vulnerability; six documents refer to all the three reasons for vulnerability.

**Table 5. Detailed guidelines and policy statements (N = 79).**

| N°* | NAME | YEAR OF PUBLICA-TION | ORGANISATION | ADOPTED |
|-----|------|------|--------------|---------|
| 12 | A Model Regulatory Program for Medical Devices: An International Guide | 2001 | Pan American Health Organization (PAHO) | Latin America and The Caribbean |
| 13 | A Proposal for the Retrospective Identification and Categorization of Older People with Functional Impairments in Scientific Studies: Recommendations of the Medication and Quality of Life in Frail Older Persons (MedQoL) Research Group | 2018 | The Society for Post-Acute and Long-Term Care Medicine | United States |
| 14 | AGS Report on Engagement Related to the NIH Inclusion Across the Lifespan Policy | 2019 | The American Geriatrics Society | United States |
| 1 | The Belmont Report. Principles and guidelines for the protection of human subjects of research | 1979 | National Commission for the Protection of Human Subjects of Biomedical and Behavioural research | United States |
| 15 | Best Practices for Protecting Privacy in Health Research | 2005 | Office of the Privacy Commissioner of Canada (OPC), Interagency Advisory Panel on Research Ethics (PRE) and Canadian Institutes of Health Research (CIHR) | Canada |
| 16 | Clinical Investigation of Medicinal Products in the Paediatric Population (E11) | 1999 | European Medicines Agency | Europe |
| 17 | Clinical Trials and Biomedical Research | 2006 | Ministry of Health Malaysia, National Institutes of Health, Medical Review and Ethics Committee (MREC), Malaysian Industry-Government Group for High Technology (MIGHT) and Academy of Sciences Malaysia (ASM) | Malaysia |
| 18 | Conducting Science in Disasters: Recommendations from the NIEHS Working Group for Special IRB Considerations in the Review of Disaster Related Research | 2016 | NIEHS Working Group | United States |
| 19 | Declaration of Helsinki | 2024 | World Medical Association | International |
| 20 | Doing No Harm and Getting It Right: Guidelines for Ethical Research with Immigrant Communities | 2013 | New Directions for Child and Adolescent Development | Spain |
| 21 | Doing the Right Thing: Outlining the DWP's approach to ethical and legal issues in social research | 2003 | Department for Work and Pensions | United Kingdom |
| 22 | Ethical Aspects of Clinical Research in Developing Countries | 2003 | European Commission, European Group on Ethics in Science and New Technologies (EGE), European Commission and Directorate-General for Research and Innovation | Europe |
| 23 | Ethical considerations for Clinical Trials on Medical Products conducted with the Paediatric Population | 2008 | Ad hoc group for the development of implementing guidelines for Directive 2001/20/EC relating to good clinical practice in the conduct of clinical trials on medicinal products for human use | Europe |
| 24 | Ethical Considerations in Biomedical HIV Prevention Trials | 2012 | UNADIS | International |
| 25 | Ethical Guidelines | 2003 | Social Research Association (SRA) | United Kingdom |
| 26 | Ethical Guidelines for Conducting Research Studies Involving Human Subjects | 2013 | Bangladesh Medical Research Council, National Research Ethics Committee | Bangladesh |
| 27 | Ethical Guidelines for Research on Human Subject in Thailand | 2007 | Forum for Ethical Review Committees in Thailand (FERCIT) | Thailand |
| 28 | Ethics Guidelines for Human Biomedical Research | 2015 | Ministry of Health (MOH) and Bioethics Advisory Committee (BAC) | Singapore |
| 29 | Ethics in clinical research: the Indian perspective | 2011 | Indian Journal of Pharmaceutical Sciences | India |

*(Continued)*

| N°* | NAME | YEAR OF PUBLICA-TION | ORGANISATION | ADOPTED |
|-----|------|------|------|------|
| 30 | Ethics in Health Research: Principles, Structures, and Processes | 2015 | Department of Health (DH) | South Africa |
| 31 | EU-Code for Ethics for Socio-Economic Research | 2004 | RESPECT project | Europe |
| 32 | Framework for Research Ethics | 2015 | Economic and Social Research Council (ESRC) | United Kingdom |
| 33 | Framework of Guidelines for Research in the Social Sciences and Humanities in Malawi | 2011 | National Commission for Science and Technology | Malawi |
| 34 | Good Clinical Practice Guidelines | 2020 | National Agency for Food, Drug Administration and Control (NAFDAC) | Nigeria |
| 35 | Good Clinical Practice Guidelines for Clinical Research in India | 2001 | Central Drugs Standard Control Organization (CDSCO) and Office of Drugs Controller General of India (DCGI) | India |
| 36 | Guidance synthesis. Medical research for and with older people in Europe: proposed ethical guidance for good clinical practice: ethical considerations | 2013 | The Journal of nutrition, health and aging | Europe |
| 37 | Guide for research ethics committee members | 2010 | Council of Europe, Steering Committee on Bioethics | Europe |
| 38 | Guide to Internet Research Ethics | 2019 | National Committee for Research Ethics in the Social Sciences and the Humanities (NESH) | Norway |
| 39 | Guideline for Application to Conduct Clinical Trials in Liberia | 2014 | Liberia Medicines and Health Products Regulatory Authority | Liberia |
| 40 | Guideline for Good Clinical Practice (GCP) in Sierra Leone | 2018 | Ministry of Health e Pharmacy Board of Sierra Leone | Sierra Leone |
| 41 | Guideline for Regulating the Conduct of Clinical Trials Using Medicines in Human Participants | 2012 | Ministry of Health and Wellness | Botswana |
| 42 | Guidelines for Conducting Clinical Trials of Medicines, Food Supplements, Vaccines, and Medical Devices in Sierra Leone | 2014 | Ministry of Health e Pharmacy Board of Sierra Leone | Sierra Leone |
| 43 | Guidelines for Good Clinical Practice E6 (and Integrated Addendums E6(R2)-(R3)) | 2025 | International Conference on Harmonization (ICH) | International |
| 44 | Guidelines for Including People with Disabilities in Research | 2002 | National Disability Authority | United Kingdom (Ireland) |
| 45 | Guidelines for Phase I Clinical Trials | 2018 | Association of the British Pharmaceutical Industry (ABPI) | United Kingdom |
| 46 | Guidelines for Research Among Children and Young People | 2011 | National Children's Bureau (NCB) | International |
| 47 | Guidelines for Research Ethics in the Social Sciences and the Humanities | 2022 | National Committee for Research Ethics in the Social Sciences and the Humanities (NESH) | Norway |
| 48 | Guidelines on Ethics for Health Research in Tanzania | 2009 | Ministry of Health (MOH), National Institute for Medical Research (NIMR), National Health Research Ethics Committee (NHREC) e Tanzania Commission for Science and Technology (COSTECH) | Tanzania |
| 49 | Guidelines on Ethics for Medical Research, Reproductive Biology and Genetic Research | 2002 | Medical Research Council of South Africa (MRC) | South Africa |
| 50 | Guidelines on Regulating the Conduct of Clinical Trials in Human Participants | 2016 | Zambia Medicines Regulatory Authority | Zambia |
| 51 | Handbook for Good Clinical Research Practice (GCP): Guidance for Implementation | 2005 | World Health Organization (WHO) | International |
| 52 | Implementing Regulations of the Law of Ethics of Research on Living Creatures | 2022 | National Committee of BioEthics | Saudi Arabia |

*(Continued)*

| N°* | NAME | YEAR OF PUBLICA-TION | ORGANISATION | ADOPTED |
|---|---|---|---|---|
| 53 | Institutional Review Board (IRB) Policies and Procedures Handbook | 2020 | University of Liberia-Pacific Institute for Research and Evaluation Institutional Review Board (UL-PIRE IRB) | Liberia |
| 54 | International Code of Marketing & Social Research Practices | 2016 | European Society for Opinion & Marketing Research (ESOMAR) | Europe |
| 55 | International Ethical Guidelines for Research Involving Humans (CIOMS) | 2016 | Council for International Organizations of Medical Sciences (CIOMS) | International |
| 56 | Malaysian Phase I Clinical Trial Guidelines | 2017 | Ministry of Health Malaysia, National Pharmaceutical Regulatory Agency (NPRA), National Committee for Clinical Research (NCRC), Clinical Research Malaysia (CRM), Ministry of Health e Society of Clinical Research Professionals Malaysia (SCRPM) | Malaysia |
| 57 | Medical Products in Human Medicine Act | 2020 | Bulgarian Drug Agency (BDA) | Bulgaria |
| 58 | Medical Research Involving Children | 2004 | Medical Research Council (MRC) | United Kingdom (England) |
| 59 | National Ethical Guidelines for Biomedical and Health Research Involving Human Participants | 2017 | Indian Council of Medical Research (ICMR) | India |
| 60 | National Ethical Guidelines for Biomedical Research Involving Children | 2017 | Indian Council of Medical Research (ICMR) | India |
| 61 | National Ethical Guidelines for Health and Health-Related Research | 2017 | Philippine Health Research Ethics Board (PHREB) | Philippines |
| 62 | National Guidelines for Ethical Conduct of Research Involving Human Subjects | 2008 | National Council for Science and Technology (NCST) e Ministry of Health (MOH) | Kenya |
| 63 | National Guidelines for Ethics Committees Reviewing Biomedical and Health Research During Covid-19 Pandemic | 2020 | Indian Council of Medical Research (ICMR) | India |
| 64 | National Guidelines for Research Involving Humans as Research Participants | 2014 | Uganda National Council for Science and Technology (UNCST) | Uganda |
| 65 | National Health Research Ethics Review Guideline, Fourth Edition | 2014 | FDRE Ministry of Science and Technology | Ethiopia |
| 66 | National Statement on Ethical Conduct in Human Research | 2018 | National Health and Medical Research Council (NHMRC) | Australia |
| 67 | Nigerian Code of Health Research Ethics | 2007 | Federal Ministry of Health | Nigeria |
| 68 | Note for guidance on Good Clinical Practice (CPMP/ICH-135/95) | 2000 | Therapeutic Goods Administration (TGA) | Australia |
| 69 | Personal Information in Biomedical Research | 2007 | Ministry of Health (MOH), Personal Data Protection Commission (PDPC) e Bioethics Advisory Committee (BAC) | Singapore |
| 70 | Policy for the Protection and Welfare of Vulnerable Adults and the Management of Allegations of Abuse | 2022 | Avista | Ireland |
| 71 | Policy Statement Regarding Enrollment of Children in Research in Nigeria | 2016 | National Health Research Ethics Committee | Nigeria |
| 72 | Qualitative methods in end-of-life research: Recommendations to enhance the protection of human subjects | 2003 | Journal of Pain and Symptom Management | California |
| 73 | Recommendation (99) 4 on principles concerning the legal protection of incapable adults | 1999 | Committee of Ministers to member states | Europe |
| 74 | Regulation (EU) 2017/745 of the European Parliament and of the Council of 5 April 2017 on medical devices | 2017 | European Parliament and Council of 5 April 2017 | Europe |
| 75 | Regulation No. 536/2014 of the European Parliament and of the Council on Clinical Trials on Medicinal Products for Human Use, Repealing Directive 2001/20/EC | 2014 | European Parliament | Europe |

*(Continued)*

**Table 5.** (Continued)

| N°* | NAME | YEAR OF PUBLICA-TION | ORGANISATION | ADOPTED |
|---|---|---|---|---|
| 76 | Regulations Relating to Research with Human Partici-pants No. R719 | 2014 | Department of Health (DH), Medical Research Council of South Africa (MRC), Human Sciences Research Council (HSRC) e South African Health Products Regulatory Authority | South Africa |
| 77 | Research Consent for Cognitively Impaired Adults. Recommendations for Institutional Review Boards and Investigators | 2004 | Alzheimer's Association | Europe |
| 78 | Research Ethics Framework (REF) | 2005 | Economic and Social Research Council (ESRC) | United Kingdom |
| 79 | Research Ethics Policy and Procedures | 2011 | University of the West Indies – Cave Hill/ Ministry of Health | Barbados |
| 80 | Research Governance Framework | 2008 | Department of Health | Bermuda |
| 81 | Resolution CNS No. 466/2012 on Guidelines and Rules for Research Involving humans Subjects | 2012 | National Health Council (CNS) and National Commission on Research Ethics (CONEP) | Brazil |
| 82 | South African Good Clinical Practice: Clinical Trial Guidelines | 2020 | Department of Health (DH) e Health Products Regulatory Authority | South Africa |
| 83 | Standards and Operational Guidance for Ethics Review of Health-Related Research with Human Participants | 2011 | World Health Organization (WHO) | Interna-tional |
| 84 | The ethics of research related healthcare in developing countries | 2002 | Nuffield Council on Bioethics | Interna-tional |
| 85 | Tri-Council Policy Statement: Ethical Conduct for Research Involving Humans | 2022 | Canadian Institutes of Health Research, Natural Sciences and Engineering Research Council of Canada and Social Sciences and Humanities Research Council of Canada | Canada |
| 86 | U.S. 45 CFR 46 | 2018 | Health and Human Services (HHS) | United States |
| 87 | Universal Declaration on Bioethics and Human Rights | 2005 | UNESCO | Interna-tional |
| 88 | Universal Declaration on Bioethics and Human Rights: perspectives from Kenya and South Africa | 2008 | Langlois, A | Kenya e South Africa |
| 89 | Updating protections for human subjects involved in research. Project on Informed Consent, Human Research Ethics Group | 1998 | Moreno, J., Caplan, A. L., Root Wolpe, P., Members of the Project on Informed Consent, Human Research Ethics Group | United States |

* Publication identification number.

The number of documents addressing Q4 is similar to the number covering Q3: seventy policy documents give some indications about the provisions to be implemented for vulnerable populations in research ethics. It is noteworthy that eight documents provide information that pertains to all four research questions.

All of the policy documents presented in our systematic review contain the root "vuln-", whereas "frag-" is mentioned in only five cases and "frail-" in eight cases (see Supporting Information section, S5 Table). This finding corroborates the idea that the correct term to be used in research ethics is "vulnerability", whereas "fragility" and "frailty" are more commonly used in clinical contexts.

Research findings are organised in four different subsections, corresponding to the four research questions reported in the "Materials and Method" Section: (Q1) meaning and definition of vulnerability as explicitly reported in research ethics policy documents; (Q2) groups/populations identified as vulnerable in research ethics policy documents; (Q3) normative justifications for vulnerability in research ethics policy documents; (Q4) consequent provisions for vulnerable populations in research ethics policy documents.

                                      

**Table 6. Distribution by continents and years.**

| CONTINENT | NUMBER OF PUBLICATIONS |
|---|---|
| Africa | 19 |
| Asia | 13 |
| Europe | 23 |
| International | 9 |
| Oceania | 2 |
| The Americas<br>North America<br>Canada<br>South America | 13<br>9<br>2<br>2 |
| **YEARS** | **NUMBER OF PUBLICATIONS** |
| 1979–2000 | 5 |
| 2001–2010 | 26 |
| 2011–today | 48 |

## Meaning and definition of vulnerability

An analysis of the ten documents that attempt to define the concept of vulnerability does not reveal homogeneity – neither in terms of the factors involved, nor in terms of what being vulnerable entails.

By closely considering the content of the policies, however, we can identify five main factors of vulnerability: i) diminished ability to safeguard one's own interests; ii) increased likelihood of incurring additional harm/risk; iii) inability to provide a valid informed consent; iv) a condition of disadvantage that depends on individual or group circumstances; v) limited decision-making capacity, broadly understood.

The following table shows how these definitions are numerically distributed across the various policy documents analysed. Although most documents only mention one of these factors as the main element defining the concept of vulnerability (Table 7), there are a few documents that correlate two factors in order to provide a definition of vulnerability.

The diminished ability to safeguard one's own interests, factor i), is the aspect that is mostly referred to when explaining the meaning of vulnerability, and policy documents provide different nuances of this definition. In *Ethics in Health Research: Principles, Structures and Process* [12], issued by the South African Department of Health (DH), it is stated that this inability "*may be caused by limited capacity or limited access to social goods like rights, opportunities and power*" [[13] p. 79], while the authors of *National Ethical Guidelines for Biomedical and Health Research Involving Human Participants* [14] argue that this incapability may be due to different reasons: personal disabilities, environmental burdens, social inequalities, lack of power, understanding or ability to communicate (or a situation that prevents one from doing so).

**Table 7. Five main definitions of vulnerability.**

| DEFINITION | NUMBER OF DOCUMENTS |
|---|---|
| i) diminished ability to safeguard one's own interests | 3 (docs n° 30, *59* and *64*) |
| ii) increased likelihood of incurring additional harm/risk | 2 (docs n° *55* and *84*) |
| iii) inability to provide a valid informed consent | 1 (doc n° *81*) |
| iv) a condition of disadvantage that depends on individual or group circumstances | 1 (doc n° *85*) |
| v) limited decision-making capacity | 1 (doc n° *61*) |

Furthermore, according to *National Guidelines for Research Involving Humans as Research Participants* [15] of the Uganda National Council for Science and Technology (UNCST), the inability to protect one's own interest is undermined by a series of conditions, such as:

*limited economic empowerment; Conflict and post-conflict situations; Inadequate protection of human rights; Discrimination on the basis of health status; Limited availability of health care and treatment options; Communities in acute disaster and disease epidemic. […] Lack of capability to give informed consent, lack of alternative means of obtaining medical care or other expensive necessities, or being a junior or subordinate member of a hierarchical group.* [[16] p. 25 and 38]

Only in one document, *Guidance synthesis. Medical research for and with older people in Europe: proposed ethical guidance for good clinical practice: ethical considerations* [17], this element of incapacity is considered as more specifically related to the problems and characteristics of certain study populations, without however defining which ones.

Two documents, [18]  and [19], describe the concept of additional harm/risk (factor ii) as the possibility of being subjected to undue influence and/or deception in clinical research. In particular, *The ethics of research related healthcare in developing countries* [20] defines vulnerability as any situation in which "*guaranteeing substantial benefits for taking part in research is more likely to constitute an undue inducement*" [[21] p. 80].

*International Ethical Guidelines for Research Involving Humans* [22], issued by the Council for International Organizations of Medical Sciences (CIOMS), adds a further element, which rejects the group-based approach mentioned in the introduction of this work: "*the account of vulnerability in this Guideline seeks to avoid considering members of entire classes of individuals as vulnerable*" [22 p. 57].

In *Personal Information in Biomedical Research* [23] a second element is introduced, associating vulnerability with two different conditions: the exposure to a greater risk of suffering negative consequences during research and the compromised ability to give voluntary consent. However, no further specification is provided for either of these items.

Factor iii) is explained in depth in the *Resolution CNS No. 466/2012 on Guidelines and Rules for Research Involving humans Subjects* [24], which defines vulnerability as "*the condition of individuals or groups that, for any reason whatsoever, have their self-judgment capability reduced or disabled, or by any means they are prevented to resist to the opposition, especially when it comes to the informed consent form*" [[25]81 p. 3].

As to factor iv), *Tri-Council Policy Statement: Ethical Conduct for Research Involving Humans*, 2nd Edition [26], defines vulnerability as a condition depending on the circumstances in which individuals and groups find themselves. This document also provides some examples of such conditions, such as a lack of rights, opportunities and power of research participants, or a social and/or legal stigmatization associated with their activity or identity. Consequently, according to this document, vulnerability can be experienced to different degrees and at different times, depending on the specific circumstances.

Finally, as to factor v), which concerns the limited decision-making capacity, it is argued in *National Ethical Guidelines for Health and Health-Related Research* [27], issued by the Philippine Health Research Ethics Board (PHREB), that this inability can be compromised by different factors: physical or mental disabilities, poverty, asymmetrical power relations and marginalisation. Individuals are not able to decide autonomously whether or not to participate in research, and this makes them vulnerable.

As our findings demonstrate, then, it is not possible to find a univocal and homogeneous definition of vulnerability in research ethics documents. At the same time, we would argue that these five defining factors, which are sometimes combined in the same document, should be considered as major elements contributing to the conceptualisation of the notion of vulnerability.

## Identification of vulnerable group/vulnerable populations

As many as seventy-two documents provided a definition of a vulnerable group/population, rendering our second research question as the one covered most often in the documents under review. This suggests that research ethics guidelines tend to associate vulnerability with specific groups and/or populations.

Despite this general trend, our analysis of the nature of the content in relation to this research question reveals a great definitional complexity, similar to what has been observed for the previous research question. Indeed, the documents that provide a definition of vulnerable group/population can be grouped into three macro-categories: (*i*) documents that define vulnerable groups a priori (e.g., children, the elderly, prisoners, etc.), therefore endorsing the group-based approach; (*ii*) documents that inferentially provide a definition of vulnerable group by considering the characteristics of vulnerable populations (e.g., those who are unable to give free and informed consent); (*iii*) documents that adopt mixed approaches.

A total of twenty documents adopt only the group-based approach in the definition of vulnerable populations (macro-category *i*.): [13,28–41] These documents feature lists of vulnerable groups/populations, without providing any further explanation. Therefore, homogeneity or common criteria are difficult to find in these lists.

The most frequently identified categories are the following: children, included in twelve documents [42,43,44,45,28,46,47,48,23,49,20,26]; people with learning disabilities or cognitive impairment [43,44,45,48,29,49], prisoners [30,43,48,50,26,51] and people with mental disabilities [30,43,44,23,52,26], included in six documents each; the elderly [44,45,28,52,49], subordinates [43,45,47,23,49] and pregnant women [30,48,52,50,51], included in five documents each; and people with serious illnesses (i.e., those with a terminal illness, people with multiple chronic conditions, physically and psychologically disabled people), included in three documents [48,52,20].

All remaining documents will either use merely the features of vulnerable populations to define them (macro category *ii*.), or they will mention some characteristics and then provide a wide range of examples (macro-category *iii*.).

These two categories of documents mostly use definitions drawn from four fundamental research ethics documents: *The Belmont Report* [1], the *Declaration of Helsinki* [53], *Guidelines for Good Clinical Practice (GCP) E6 and Integrated Addendums E6(R3)* [54] and *International Ethical Guidelines for Research Involving Humans (CIOMS)* [22]. The *Declaration of Helsinki* belongs to macro-category *ii*., while the other three documents adopt mixed approaches and therefore belong to the third macro-category. Our analysis of macro-categories *ii*. and *iii*. actually started from these four fundamental documents.

The *Declaration of Helsinki* (2013) [53], which uses the features of vulnerable populations to define them, introduces a distinctive element: it recognises the existence of groups and individuals with different degrees of vulnerability, affirming the existence of subjects who can be considered "particularly vulnerable/more vulnerable", compared to others, "*due to factors that may be fixed or contextual and dynamic, and thus are at greater risk of being wronged or incurring harm*" [21] p. 3].

Three documents [31,55,56] feature this same definition of "particularly/extremely vulnerable'" subjects, even if each of them provides its own lists of vulnerable subjects.

Turning to macro-category *iii*., *The Belmont Report* [1], the oldest of the four foundational documents, first identifies some groups as vulnerable in research (i.e., racial minorities, economically disadvantaged people, the very sick and the institutionalized), after which it attributes their vulnerability to some general characteristics, namely "*their dependent status and their frequently compromised capacity for free consent*" [[1] p. 9].

GCP Guidelines [54] provides yet another definition of vulnerable group/population, which was subsequently adopted in a significant number of later policy documents. This definition highlights the link between vulnerability and undue influence, stating that vulnerable subjects might be "*unduly influenced by the expectation, whether justified or not, of benefits associated with participation, or of a retaliatory response from senior members of a hierarchy in case of refusal to participate*" [54] p. 78].

In the *GCP Guidelines*, this definition is followed by a list of individuals identified as vulnerable (i.e., members of a group with a hierarchical structure, people in nursing homes, the unemployed, poor people, patients in emergency situations, ethnic minority groups, the homeless, nomads, refugees, minors and people incapable of giving consent).

Six policy documents mention the same identical definition of vulnerable group as the one provided by GCP Guidelines: [18,57,58,59,32,13]. Another two documents, after having reported the definition textually, add some extra parts: *Guideline for Good Clinical Practice (GCP) in Sierra Leone* [60] includes in its list of vulnerable subjects also those who are unconscious; *South African Good Clinical Practice: Clinical Trial Guidelines* [61], instead, mentions a range of factors that can worsen their vulnerability:

> *diminished health, loss of liberty or other health-related personal circumstances, including adults with diminished decisional capacity, persons with mental illness, mental disability, or persons who have substance abuse problems, persons in dependent relationships, incarcerated offenders and persons highly dependent on medical care.* [61 p. 15]

The last of the four fundamental documents, CIOMS Guidelines [22], draws attention to two characteristics that are key in defining groups as vulnerable: the inability to protect one's own interests (due to a lack of decisional capacity, education, resources, strength or due to circumstances of living), and a limited capacity to consent (or refuse consent). Each of these two elements is complemented by an extensive list of examples.

As to the first, characteristic examples are: individuals who face social exclusion or prejudice, illiterate people and people living in an authoritarian environment. As to the second characteristic, the document mentions individuals in hierarchical relationships, institutionalised persons, homeless persons, refugees or displaced persons, people living with disabilities, etc.

A good share of documents in macro-category *iii* uses the definitions provided by the abovementioned fundamental documents to justify the identified vulnerable groups, without quoting them literally.

In particular, ten documents [62,63,64,65,16,66,67,15,68,69] base the reasons of the vulnerability of their groups on the inability to provide free informed consent [cf. [1] and [18]. One of these, *Ethical Guidelines for Research on Human Subject in Thailand* [16], adds to the definition of vulnerable population also those "*who have inferior or lack physical capacities or have diminished capacities for making a reasonable decision*" [[29] p. 3].

*National Guidelines for Research Involving Humans as Research Participants* [15], instead, adds some groups which are considered vulnerable a priori, without providing any further explanation for it: the economically disadvantaged, people in conflict or post-conflict situations, people whose human rights are violated, people stigmatised because of their health status, people without access to health care and treatment, and people suffering from disasters or disease outbreaks. *Policy for the Protection and Welfare of Vulnerable Adults and the Management of Allegations of Abuse* [68] integrates the definition by mentioning the risk of abuse, harm and exploitation that also characterizes vulnerable populations.

Seven documents [33,70,71,72,32,73,74] define vulnerable individuals as those who are exposed to undue influence (referencing GCP Guidelines), while also offering a number of different examples, if without adding any further specifications. For example, Institutional Review Board (IRB) Policies and Procedures Handbook [72] only mentions these categories as vulnerable subjects: "*children, prisoners, individuals with impaired decision-making capacity, or economically or educationally disadvantaged persons*" [72 p. 20].

Three documents [75,14,76] mention the inability to protect one's own interests, as outlined in the CIOMS Guidelines [22], but they list different examples. *National Ethical Guidelines for Biomedical and Health Research Involving Human Participants* [14] further explains that individuals may be vulnerable due to elements such as legal status, clinical conditions, situational conditions and increased psychological, social, physical or legal risks, while *Regulation Relating to Research with Human Participants No. R719* [76], issued by the Department of Health (DH), Medical Research Council of South Africa (MRC), Human Sciences Research Council (HSRC) and South African Health Products Regulatory Authority,

inferentially paraphrases who the vulnerable subjects are (e.g., those who are exposed to increased risk of harm during research and those who are restricted in their freedom of choice).

Finally, ten documents define vulnerable groups/populations by combining different definitions: *A Model Regulatory Program for Medical Devices: An International Guide* [77] refers to both the *Declaration of Helsinki* [53] and *The Belmont Report* [1]; *Ethical Considerations in Biomedical HIV Prevention Trials* [34] combines the GCP Guidelines [54] with the *Declaration of Helsinki* [53] (in this guideline, the definition of vulnerable group is also based on stigma, discrimination and marginalisation, as it refers to biomedical HIV prevention trials, providing a number of meaningful examples); documents [78] and [27] refer to both GCP Guidelines [54] and *The Belmont Report* [1] (document [27] has been included in this category even though it does not state the textual words "consent" and "undue influence". Indeed, the document mentions "persons most susceptible to coercion" and "persons relatively or completely incapable of deciding for themselves whether or not to participate in a study", but the assumed meaning is assimilable); documents [35] and [24] combine definitions from GCP Guidelines [54] and CIOMS Guidelines [22]; *National Ethical Guidelines for Biomedical Research Involving Children* [79] draws definitions from the *Declaration of Helsinki* [53], the *Belmont Report* [1] and GCP Guidelines [54]; documents [80,36] and [81] refer to CIOMS Guidelines [22], GCP Guidelines [54] and the *Belmont Report* [1].

Seven documents differ from all the previously mentioned ones because they provide an original definition of vulnerable group/population – one that does not clearly reference any of the already covered elements. More specifically, documents [12] and [82] state that the vulnerable subjects are those who are unable, or less able, to understand information, while, according to documents [83,25,37,84], vulnerability is connected to an impaired autonomy in decision-making process.

The last of these seven documents, *Implementing Regulations of the Law of Ethics of Research on Living Creatures* [38], defines vulnerable groups as groups of individuals who lack legal capacity and have dubious, reduced or absent ability or freedom to choose.

The analysis of these findings pertaining to our second research question clarified that policy documents more frequently present a definition of vulnerable groups/populations, rather than one of the concept of vulnerability as such. Regardless of this fact, there is still much terminological difficulty, as well as a lack of homogeneity in the groups identified as vulnerable.

## Normative justifications for vulnerability

As stated in the introduction, the normative justifications for vulnerability can be classified into three major categories: "consent-based reasons" for vulnerability, "harm-based reasons" for vulnerability and "justice-based reasons" for vulnerability. The justifications reported in the analysed documents were traced back to these three macro-categories only when explicitly inferable. The following section will provide information concerning the specific statements, in the reviewed documents, linked to the different sources of vulnerability.

In the category "consent-based reasons" for vulnerability, we observed several aspects, such as, but not limited to, the inability to give a free and informed consent, the possibility of suffering pressure or coercion during consent procedures or during research, the fear of retaliation in case of refusal to participate, and the possibility of being easily manipulated by someone who has a need for easily recruitable research subjects.

Broadly speaking, all the elements associated with this first category related to the (in)ability to make free and informed decisions about participation in research.

In the category "harm-based reasons" for vulnerability, we included all documents which mentioned as a key issue the increased likelihood of being wronged or of incurring additional harm during the research. These documents refer to vulnerable individuals as incapable of protecting their own interests and as individuals exposed to undue risk. The reason for vulnerability might also be "harm-based" due to certain environmental and contextual situations that increase the risk of harm.

 

Finally, the category "justice-based reasons" for vulnerability refers to documents in which the vulnerable subjects are described as those who cannot benefit from an adequate distribution of resources and services and are, therefore, disadvantaged. Examples of vulnerability resulting from injustice are situations of poverty, limited availability of healthcare resources and treatment options, individuals living with incurable conditions, individuals who are politically powerless, and individuals not allowed to benefit from the results of the research in which they participate.

The analysis of the documents under review shows that most (but not all) policy documents address the normative justifications for vulnerability: sixty-eight of the total number of documents included in the review provide some reasons for vulnerability, accounting for about 86% of the total.

Considering that some of the documents under review attribute vulnerability only to one of the three identified categories of justifications, while others combine more than one reason for vulnerability, the various justifications are distributed across the various documents as follows: sixty-five documents mention a "consent-based reason" for vulnerability, thirty documents mention a "harm-based reason" for vulnerability, while twelve documents mention a "justice-based reason" for vulnerability.

More than a single normative justification for vulnerability is present in a significant number of documents: twenty-one documents mention both "consent-based" and "harm-based" reasons for vulnerability [19,42,43,16,75,17,78,35,14,27,36,47,67,79,81,23,68,69,76,21,20], six documents mention both "consent-based" and "justice-based" reasons for vulnerability [1,19,42,45,61,26], and six documents refer to all the three reasons for vulnerability [33,34,12,80,22,67].

When considered in more specific detail, a clear majority of the documents refers to consent-based reasons for vulnerability, strongly based on a logic in which vulnerable subjects feature as those whose decision-making ability and consent-giving capacity are compromised. This majority appears to be clear when considering the documents reporting "consent-based" as the only normative justification of vulnerability (thirty-two documents [31,63,64,65,70,18,66,83,28,55,57,58,60,54,71,85,86,59,25,38,72,32,39,82,13,37,24,49,50,84,73,74]) as well as those that use more than one normative justification (the linking of consent-based and harm-based reasons being the most common).

Harm-based reasons are the second most frequently mentioned justifications in the documents under review. However, as reported, harm is found as the only justification in just three documents [40,53,51]. Twenty-one documents join consent-based and harm-based reasons.

No document solely cites a normative justification grounded in justice. A "justice-based reason" for vulnerability can be found only in the six documents that present all three justifications and in the six documents that combine justice-based reasons with consent-based reasons.

## Provisions for vulnerable populations

Policy documents often stress the necessity to identify appropriate provisions in order to meet the needs of vulnerable participants. Seventy of the reviewed documents addressed our fourth research question, while most documents comprise more than one provision.

In analysing this data, we classified the provisions into two macro-categories: "first-level provisions" and "second-level provisions". Under the definition of "first-level provisions", we inserted all the general research ethics provisions, namely:the essential requirements applicable to all potential participants, vulnerable as well as non-vulnerable, even if, in the documents reviewed, they pertain to provisions that refer specifically to a vulnerable population (see Table 8).

The aspect on which these provisions most frequently rely is the informed consent process: fifteen documents state that the participation of vulnerable subjects in research must be determined by a free and informed choice [31,65,16,70,12,45,83,17,86,38,25,27,82,76,49]. This implies informing the participants about the rationale of the study, its

**Table 8. First-level provisions.**

| PROVISIONS | NUMBER OF DOCUMENTS |
|---|---|
| Participation determined by a free and informed choice | 15 (docs n° *21, 25, 27, 28,* 30, 32, *33, 36,* 47, *52, 54, 61, 66, 76* and *78*) |
| Demonstrate the appropriateness of the inclusion criteria adopted in the research protocol | 13 (docs n° *1, 12, 27, 29,* 30, *35–37, 42, 59, 63, 65* and *75*) |
| Fair distribution of burdens and benefit | 3 (docs n° *35, 58* and *85*) |
| Grant the right to confidentiality of information | 3 (docs n° *24, 32* and *69*) |
| People should benefit from the knowledge, practices and interventions resulting from research | 2 (docs n° *75* and *80*) |
| Ability to provide consent assessed on the basis of the personal characteristics of each individual | 1 (doc n° *47*) |

implications and results in a way that is the most appropriate to the subject's ability to understand. In this manner, individuals can actively choose whether to participate in the study.

One document, *Guidelines for Research Ethics in the Social Sciences and the Humanities* [86], issued by the National Committee for Research Ethics in the Social Sciences and the Humanities (NESH), specifies that the ability to provide consent "*should be evaluated based on individual competence, not on group characteristics*" [[49] p. 29].

Furthermore, in order to ensure adequate inclusion of all potential participants in the study, thirteen guidelines state that it is necessary to demonstrate the appropriateness of the inclusion criteria adopted in the research protocol [1,77,16,75,12,17,55,66,78,14,36,81,52]. Three documents [34,83,23] stress the importance to grant the right to confidentiality of information and three other policies consider a fair distribution of burdens and benefits as essential [66,47,26]. For instance, in *Good Clinical Practice Guidelines for Clinical Research in India* [66] it is stated that "*effort may be made to ensure that individuals or communities invited for research be selected in such a way that the burdens and benefits of the research are equally distributed*" [[37] p. 41].

Lastly, two guidelines [52 and 84] claim that research is only justifiable if it enables vulnerable people to benefit from the resulting practices and interventions and when it is "*intended to develop knowledge with the prospect of delivering health-related benefits for that particular population*" [[81] p. 7].

In addition to this first set of general provisions, a good share of guidelines will also provide several specific provisions, targeted to vulnerable individuals. We identified these provisions as "second-level provisions", after which we further classified them into two sub-categories: "broad-grained provisions" and "fine-grained provisions".

**Table 9. Broad-grained second-level provisions.**

| PROVISIONS | NUMBER OF DOCUMENTS |
|---|---|
| Introduction of specific protections to safeguard the rights, safety and well-being of vulnerable participants | 35 (docs n° *16, 17, 19,* 20, 22, *25–27, 30, 31, 34, 35, 38, 40, 42, 43, 47, 48, 51, 53, 55, 59–61, 63, 64, 67, 73, 79,* 80, *85–89*) |
| Vulnerable people not exploited | 11 (docs n° *17, 24, 48, 49, 52, 59, 62, 70, 71, 84* and *88*) |
| Justify the exclusion of vulnerable subjects | 7 (docs n° *17, 37, 55, 67, 69, 76* and *85*) |
| Consider all factors contributing to vulnerability beforehand | 7 (docs n° *24, 26, 27, 48, 55, 59* and *85*) |
| Avoid an excessive and systematic exclusion | 3 (docs n° *18, 37* and *55*) |
| Exclusion under special conditions | 2 (docs n° *38* and *40*) |

By "broad-grained (second-level) provisions" we mean general requirements that must be fulfilled specifically for vulnerable people (see Table 9).

In this context, but also in general with regard to the fourth research question, the most frequently mentioned provision is the introduction, in the research setting, of specific protections to safeguard the rights, safety and well-being of vulnerable participants.

This provision is found in as many as thirty-five documents, but in most of them it is not made explicit what these protective measures consist of and how they should be implemented [42,30,53,33,63,16,43,65,12,44,18,66,28,60,78,54,86,35,80,72,22,14,27,79,36,15,48,37,50,84,26,51,73,74,87].

The same vagueness can be found in relation to the importance of avoiding the exploitation of vulnerable people. In fact, eleven guidelines [30,34,35,88,38,14,67,68,69,20,51] stress this issue, but specific indications to prevent the exploitation are not suggested.

Instead, a provision that can be traced back to the very early years of the discussion on clinical research ethics is the exclusion of vulnerable subjects from research. Interestingly, however, our systematic review found only two documents that mention the exclusion of vulnerable participants, under special conditions: for instance, "*when the information is especially sensitive and the informants are vulnerable*" (*Guide to Internet Research Ethics* [[40] p. 15]) and when the study is carried out with unconscious subjects and/or individuals in emergency situations (*Guideline for Good Clinical Practice (GCP) in Sierra Leone* [60]). According to seven other documents, any decision to exclude a vulnerable person must be justified [30,55,22,48,23,76,26], while two of these [55 and 22] emphasise the need to avoid excessive and systematic exclusion. This latter provision, in a slightly different context, is also mentioned in *Conducting Science in Disasters: Recommendations from the NIEHS Working Group for Special IRB Considerations in the Review of Disaster Related Research* [40].

According to all the other documents, vulnerable people may, and should, be included in research, but with specific precautions and safeguards.

The last provision included in this broad-grained second-level category recommends that all factors contributing to vulnerability that may exacerbate it in the research context, should be appropriately taken into consideration beforehand, as can be identified in seven policies [34,43,16,35,22,14,26].

Finally, we defined "fine-grained (second-level) provisions" as those provisions that can be considered practical, operational indications on how to deal with vulnerable subjects during the course of the trial (see Table 10).

In this context, the theme of consent provided by proxy or by an authorised legal representative is addressed. A total of eighteen guidelines deal with this topic. Of these, thirteen documents state, in general, that proxy consent or consent provided by a legal guardian is necessary and should be sought when subjects are unable to consent first-hand [31,63,43,16,45,17,54,80,25,22,82,23,84]. In most cases, no explanation of why the subject is deemed unable to consent is provided.

Five documents address this issue referring particularly to the population of children and identifying them as still incapable of completely deciding for themselves due to their age and/or lack of maturity [70,45,46,14,79]. It is argued in these documents that research involving minors can only be considered ethically justifiable when there are tangible benefits for the child and when parental consent has been sought. The child's consent, however, must be obtained whenever possible, while any refusal on the part of the child must be respected.

The document on *Ethics Guidelines for Human Biomedical Research* [70], issued by the Ministry of Health (MOH) and Bioethics Advisory Committee (BAC) of Singapore, is the only one to focus on societies in which proxy consent is culturally widespread: in this case, "*while local customs are to be respected, they cannot supersede a requirement for individual consent*" [70 p. 29].

Further operational guidance can be found in twelve documents [41,75,12,46,38,22,79,15,81,48,73,74], arguing that research ethics committees should involve at least one member who has experience in working with vulnerable people and/or their advocates.

**Table 10. Fine-grained second-level provisions.**

| PROVISIONS | NUMBER OF DOCUMENTS |
|---|---|
| Research on vulnerable subjects only when it cannot be conducted on non-vulnerable individuals, and when it responds to their specific needs | 15 (docs n° *16*, *19*, *22*, *24*, *30*, *55*, *58–61*, *64*, *65*, *76*, *81* and *82*) |
| Proxy consent or consent provided by a legal guardian necessary when subjects are unable to consent first-hand | 13 (docs n° *21*, *22*, *26*, *27*, *32*, *36*,*43*, *51*, *54*, *55*, *66*, *69* and *80*) |
| Involve at least one member who has experience in working with vulnerable people and/or their advocate in the ethical committee | 12 (docs n° *14*, *29*, 30, *46*, *52*, *55*, 60, *64*, *65*, *67*, *86* and *89*) |
| Research on vulnerable individuals only if it can be conducted in the same way on legally competent individuals | 6 (docs n° *23*, *38*, *48*, 69, *81* and *82*) |
| No more than minimal risk | 6 (docs n° *27*, *48*, *55*, *66*, *78* and *85*) |
| Research involving minors only when there are tangible benefits for the child and when parental consent has been sought | 5 (docs n° *28*, *32*, *46*, *59* and *60*) |
| Constantly monitor the study and its effects | 5 (docs n° *24*, *51*, *77*, *82* and *88*) |
| The least vulnerable of a specific category subjects should be included | 3 (docs n° *26*, *37* and *48*) |
| Respect the tradition of societies where proxy consent is culturally widespread, but requiring individual consent | 1 (doc n° *28*) |

The recruitment of vulnerable subjects is another issue to which these provisions pertain. In general, most guidelines address the recruitment issue by asserting that research with vulnerable individuals is ethically acceptable only when it cannot be conducted on non-vulnerable individuals, and when it responds to their specific needs. This provision is included in fifteen documents [42,53,63,34,12,22,14,27,47,79.15,81,76,24,61]. More specifically, according to six documents, research involving vulnerable individuals is not justifiable if it can be conducted in the same way on legally competent individuals [64,28,35,23,24,61]. For instance, in *Guidelines on Ethics for Health Research in Tanzania* [35] it is claimed that "*research that can be carried out on subjects who can consent should not be carried out on individuals who have no capacity to understand, or freedom to refuse is limited*" [35 p. 47].

Three policies argue that, if it is indispensable to recruit vulnerable subjects, the least vulnerable from within that category should be included [43,55,35].

Furthermore, no more than minimal risk must be allowed, according to documents [16,35,22,82,49,26], to ensure ethically acceptable research.

Finally, five guidelines [34,80,29,61,51] articulate the importance of sustained monitoring of the study and its effects. For example, the *Handbook for Good Clinical Research Practice (GCP): Guidance for Implementation* of the World Health Organization (WHO) [80] claims that safeguards may include "*seeking permission of a legal guardian or other legally authorized representative when the prospective subject is otherwise substantially unable to give informed consent; […] and/or additional monitoring of the conduct of the study*" [[80]p. 22].

## Discussion

This systematic review allowed us to gain a comprehensive overview of the concept of vulnerability in research ethics as it has been reported within policy documents and guidelines. Our analysis of a significant number of documents and their variety allowed us, as scholars with an expertise in bioethics, research ethics, biotechnology, and clinical trial coordination,

to identify some recurring themes and patterns that characterise the way vulnerability is described in and addressed by policy documents. The following sections will discuss some of the most relevant issues resulting from the above-presented results. Naturally, whilst every effort has been undertaken to refrain from overinterpreting the experimental findings, the delineation of what is herein referred to as an emerging theme is, to some extent inevitably, informed by the authors' own scholarly expertise.

### Focus on vulnerable populations and groups

First, it is noteworthy to report that research ethics documents and guidelines are mostly geared to identifying and categorising vulnerable subjects, rather than providing a conceptualisation of the notion of vulnerability as such, as attested to by the fact that only ten documents attempt to provide a definition of vulnerability.

This aspect, which does not seem to equally apply to other domains of applied ethics (in clinical ethics, for instance, a debate over the notion of vulnerability is instead prevalent [89] – see next Section), may be explained by an underlying pragmatic stance – rather than a theoretical one – characterizing the field of research ethics, which originated from the need to ensure the protection of human subjects in medical research. Although such a stance may be pragmatically intelligible, it remains far from established that endorsing what has been defined as labelling approach (see Introduction) better serves this objective (see next section).

Another, interrelated aspect emerging from the collected documents is the tendency to define particular categories of subjects as vulnerable without having defined vulnerability and its various implications. As an example, twenty documents a priori identify vulnerable subjects, without providing any further explanation and/or justification for the decision process involved. Accordingly, subjects (groups or individuals) are labelled as vulnerable *only* insofar as they belong to populations that have traditionally been considered as such. A typical example of the latter is represented by people affected by disability. People with disabilities are considered *by default* vulnerable since are defined as those "*experiencing, at any point across their lifespan, long or short-term impairments in one or more body structures or functions*" [90 p. *4*] and, therefore, at greater risk of being wronged. Clarifying the disability-vulnerability connection may be important insofar as the failure to recognise a disability makes the individual even more vulnerable, increasing the social disadvantage she suffers. However, the use of vulnerability as a legally relevant benchmark in this field can lead to situations where individuals perceive the need to be classified as vulnerable in order to have access to particular forms of protection [91]. Therefore, the language of vulnerability in the context of disability discourses may eventually end up appearing "*disempowering and objectifying*" [[92]9p. 1583].

The fact that the definitional method or labelling approach is still very common in policy documents may in part explain why a shared definition of vulnerability in research ethics is still missing.

In turn, the absence of a common understanding of what vulnerability amounts to and implies in research ethics may also explain the heterogeneity, across documents, in identifying vulnerable groups/populations, as each document reports its own explanations and, consequently, the groups seen as vulnerable vary. Even if the same group is identified as vulnerable, the explanation for its vulnerability turns out to vary.

For example, children may be seen as vulnerable because they are exposed to undue influence or because they lack the capacity to give free consent.

In spite of these differences, it should also be acknowledged that some populations are recurrently classified as vulnerable, such as children, prisoners, people with mental disabilities, the elderly, the subordinates, pregnant woman and people with serious illnesses.

Accordingly, this list may represent a homogenous core for policy documents, provided that a more explicit clarification of their common framing as vulnerable is reported as well: their vulnerability ought to be established and justified, rather than merely be linked to preconceived categories. Indeed, it is necessary "*to provide an analysis of vulnerability that does not render it vacuous, rescues its force and importance*" [[93]p. 8] (see next paragraph).

## Vulnerability in research ethics and other domains of applied ethics

A second emerging concern is that the notion of vulnerability in research ethics differs from the one provided in other contexts, as it entails a higher degree of complexity.

In other fields of applied ethics (e.g., clinical ethics), vulnerability actually tends to be considered as a defining ontological feature of human beings, who are exposed to finitude and subjected to the consequences of human embodiment, or vulnerability is considered as a situational characteristic, related to contextual factors (mostly social, political and economic) which may further worsen a personal condition [89].

In research ethics, it seems that a univocal and comprehensive definition of vulnerability cannot be achieved. The analysis of the documents which address our first research question seems to reveal a highly heterogeneous notion of vulnerability, which represents not only typical elements of the etymological definition of vulnerability (e.g., increased risk of incurring additional harm or risk), but also specific elements related to the context of research ethics (e.g., inability to provide free informed consent).

Indeed, the lack of a univocal and comprehensive definition of vulnerability may be linked to the fact that, in research ethics, vulnerability is not generally considered *per se*, but mostly within the specific context of clinical research, in relation to defined situations that may create (additional) vulnerability. In other words, subjects in clinical research may be deemed more vulnerable, not on the basis of their general characteristics or the attributes of their human nature but because they are considered, in that specific context, unable to provide free informed consent, not fully capable to protect their interests, more susceptible to undue influence, etc.

The result of this reasoning is, in our view, an incomplete understanding of the complexity of the notion, which eventually results in a procedural stance rather than a substantive concept [92]. This consideration is also in line with contemporary bioethical literature, which has extensively criticised the labelling approach, insofar as it may appear at the same time as too broad and too narrow [93]. The labelling approach has been criticised for being overly broad, insofar as, by encompassing categories of individuals rather than specific persons, it risks producing undue generalisations that may prove pragmatically ineffective. If, for instance, all elderly individuals are presumed to be vulnerable, overly cautious measures may be implemented for those who, in the specific context, do not require them—potentially resulting in outcomes ranging from stigmatisation and discrimination to exclusion from research altogether. Moreover, given that, as previously discussed, there exist numerous categories of potentially vulnerable individuals, such an approach may ultimately lack practical efficacy. Conversely, the labelling approach has also been characterised as overly narrow, insofar as it tends to adopt a somewhat reductionist perspective—one that focuses almost exclusively on concepts such as capacity and competence, and thus primarily reframes vulnerability in terms of an individual's ability (or inability) to provide ethically valid informed consent (see next section).

A further observation on the notion of vulnerability presented in the reviewed documents is that vulnerability is always strongly related to safeguarding the ethical principles of clinical research, but the respect for vulnerability is hardly ever considered a foundational concern of ethical research in its own right. As pointed out in earlier studies [such as [94], vulnerability, in this way, only serves as an indicator of other research ethics concerns, already captured by existing concepts, such as harm or consent.

In this regard, however, the question arises whether vulnerability, in and of itself, would be a more useful, effective tool for stakeholders engaged in the regulation and conduct of research. Although this issue does not constitute the primary object of inquiry in the present work, it is nevertheless noteworthy to point out that several scholars have sought to engage with it, showing how the concept of vulnerability may serve as a valuable lens through which to reconsider certain morally salient aspects of our humanity [95]. Moreover, it offers a means for addressing significant ethical concerns in a more holistic manner—"*considerations that can readily be obscured by a procedural focus on informed consent or balancing research benefits and burdens*" [93] p. 25].

A possible way for partially overcoming these difficulties would be for research ethics to borrow some of the identifying categories of vulnerability as presented in clinical ethics and philosophical bioethics debates. As some distinguished scholars have already pointed out [94; 96], promoting a less reductionist view of vulnerability would mean interpreting the former not only in relation to the violations of the principles of autonomy, beneficence, and justice, but also to recognize *layers* [94], *sources* [92], or *circumstances* [97] of vulnerability, which may be both basic human (i.e., related to the very essence of the human nature) and situational, essential and dynamic [93]. In our view, this complexity has been already anticipated by the last version of the *Declaration of Helsinki* [53] which has stressed the concept of "particular vulnerability" and the distinction between factors of "fixed" and "dynamic" vulnerability.

### Informed consent as the defining element for identifying vulnerability in research ethics

Despite the great heterogeneity observed, the analysis of the results of the present systematic review revealed an emerging and recurring theme related to vulnerability in research ethics policy documents: informed consent. Indeed, the issue of consent features prominently in the results of three out of the four research questions: categories of individuals are often identified as vulnerable because they lack the capacity to provide a valid informed consent; one of the normative justifications for their individuals' vulnerability is consent-based; and, finally, most of the provisions to comply with vulnerable research subjects are based on promoting appropriate consent practices (i.e., including the use of proxy consent or consent provided by an authorised advocate, when necessary).

Moreover, insofar as two identifying factors of vulnerability are inability to provide a valid informed consent (condition iii) and limited decision-making capacity (condition v), consent-related definitions also return in the results of the first research question.

The emphasis on consent is clearly in line with the academic debates within research ethics. Indeed, informed consent should certainly be identified as a recurring concern as well as a major value in research ethics literature and guidelines.

Moreover, there is a longstanding narrative (originating from *The Belmont Report*) which relates vulnerability to the principle of respect for persons and which is subsequently reformulated as principle of autonomy and further operationalised in the informed consent process. A disproportionate emphasis on informed consent in research ethics debates around vulnerability has also been pointed out by previous academic literature [98]. However, if we focus on selected documents, the arguments in favour of this take either lack transparency or fail to be properly justified.

A possible explanation for this phenomenon may be that precisely the lack of sustained theoretical reflection on the concept of vulnerability in research ethics may have made it more difficult to find a *file rouge* in the identification of vulnerable subjects. Within this scenario, the issue of consent may have appeared as the most tangible aspect, defined in itself, which does not require a preliminary in-depth investigation of the notion of vulnerability. According to this explanation, consent becomes both the means and the reason for defining a subject or a group of subjects as vulnerable, without necessarily having to define vulnerability first.

### Useful provisions?

Our fourth consideration emerging from the results of this systematic review pertains to the provisions presented in the various documents.

In general, the provisions provided are mostly procedural ones, such as including a member on research ethics committees who has experience in working with vulnerable individuals, enrol subjects legally able to provide consent, support vulnerable subjects in their decision-making process, etc.

Although these kinds of provisions are certainly less abstract than theoretical principles (e.g., promote autonomy), these provisions may still appear to be either not very useful or sufficiently practical for experimenters [3]. Even those

provisions that seem more operational actually fail to address how to deal effectively with vulnerable subjects during research, how to facilitate the informed consent process, how to implement any additional protective measures for vulnerable subjects. In other words, the analysed documents do not contain provisions that seem specifically intended to provide practical guidance to investigators dealing with vulnerable subjects in different research settings.

When guidance documents do not provide clear, contextual and univocal indications, it becomes very difficult for both research ethics committees and researchers to take firm and unambiguous decisions related to the treatment of vulnerable populations in research, not subjected to potential objections. In other words, within such a regulatory grey area, it is often left to the responsibility of researchers and research ethics committees to adopt caution and further interpret these (still too general) considerations when dealing with vulnerable subjects.

### Non-binding guidelines

An element which further complicates the implementation of effective provisions in clinical research is the legal nature of the guidelines we reviewed. While conducting this systematic review, we attempted to classify the reviewed policies into legally binding or legally non-binding documents, based on explicit formulations in the texts of the documents. As a result of this classification, we can consider twenty-three of the reviewed documents as legally binding [12,57,60,78,86,35,38,25,32,14,27,15,48,81,82,68,52,56,76,24,61,26,73], while forty-three are legally non-binding [1,19,41,62,77,30,40,53,31,34,63–65,16,70,75,44,45,83,17,28,55,66,58,54,71,85,59,80,88,72,22,47,79,67,36,13,23,37,69,99,29,49,50,21,20,51,74,87]. In the remaining seven documents [42,33,43,18,46,39,84], it proved impossible to decipher the legal status, as no explicit indication or reference to it was found in the text.

Accordingly, the vast majority of the documents regulating the conduct of researchers towards vulnerable populations in clinical research are non-binding guidelines, i.e., documents that provide indications, not stringent rules. The result is that even those policies that seek to be more practical and provision-oriented contribute little when it comes to establishing a course of action in concrete cases.

Moreover, even among the provisions contained in the legally binding documents, very few are operational; an exception is represented by the one requiring that research ethics committees have at least one person with experience in working with vulnerable persons and/or their legal representative [12,38,48,73]. The most frequently mentioned provision in the legally binding documents is to provide special protections for the rights, safety and welfare of individuals [12,60,14,48,73]. Also in this case, however, no further specification on how such indication should be implemented or addressed is provided.

Although most documents in our study do not result in legally binding guidelines, and therefore they run the risk of being unable to impact on the research practice, we believe that this problem can be tackled by invoking the policy-making role of research ethics committees. Today, as shown by the academic literature, research ethics committees do represent policymaking bodies, capable of acting *de facto* as research policymakers for their institutions, by interpreting indications and recommendations, as well as creating policies to navigate grey zones [100]. A potentially useful tool which may accompany research ethics committees in this task may be represented by what we labelled as "Vulnerability Checklist" (see the Supporting Information section, S1 Checklist), namely an easy-to-compile checklist that experimenters would be asked to fill in and submit to research ethics committees along with study protocols and other relevant documentation for ethics review. The checklist is structured into two main parts: first, a summary of what is meant by vulnerability in research ethics, the main reasons and conditions for vulnerability in research, as well as examples of vulnerable groups; second, a form were experimenters are asked to identify whether the proposed study enrols vulnerable subjects and/or creates conditions where vulnerability may arise, and the measures that the experimenter has envisaged to mitigate those vulnerabilities (if at all possible).

It is important to point out that proposed checklist has been developed on the basis of the results of the present systematic review. In other words, the collected evidence has informed, on a multiple level, the content of the checklist itself.

First, since our systematic review revealed that there is a combination of documents which inspired most policy documents and guidelines (namely: the Belmont Report, the Declaration of Helsinki, CIOMS Guidelines, and GCP Guidelines), checklist provisions were developed to be compliant with the statements and suggestions provided within these fundamental documents. For instance, the definition of vulnerability as a fixed/inherent and situational/dynamic notion (first part of the checklist, page 1), as well as the provisions for including vulnerable populations in research (first part of the checklist, page 2) take inspiration from the latest version of the Declaration of Helsinki, and the bioethical literature on the topic; the statement according to which "Vulnerability per se does not represent an ethically justifiable reason for excluding entire groups of individuals from research" is inspired by the 2016 version of the CIOMS Guidelines (but also from other documents, e.g., n. 17 and n. 37); the tripartite framework of consent-based, harm-based, and justice-based approaches draws inspiration not only from the bioethical literature on the subject, but also from the Belmont Report, which emphasized that vulnerability should be understood as arising from violations not solely of the principle of autonomy, but also of beneficence and justice.

Moreover, as to the list of "typical categories of potentially vulnerable individuals/groups" (first part of the checklist, page 2), the *types* of categories indicated as well as the *order* in which these typical categories are listed follow from the findings of our review: children appear as the mostly recurrent category mentioned in our included documents, followed by people with learning disabilities or cognitive impairments, people with mental disabilities, etc. (see Results, subsection "Identification of vulnerable group/vulnerable populations").

Finally, the second part of the checklist ("Section that has to be filled in by the experimenter"), is also structured in line with what we defined "fine-grained (second-level) provisions", namely practical, operational indications, reported in our included policy documents, meant to provide experimenters with some guidance to properly deal with vulnerable subjects during the course of the trial.

Although we agree with scholars claiming that vulnerability-sensitive efforts should not be restricted to the enrolment process, and that more comprehensive processes to address the entire lifecycle of research should be developed [96,98,101], we also contend that vulnerability-sensitive efforts should be at the same time useful and feasible, namely, able to offer additional safeguards for vulnerable populations beyond the ethical considerations typically detailed in standard research protocols, while, at the same time, not placing additional bureaucratic burdens upon experimenters and research ethics committees' members. In particular, we argue that the checklist proposed here scores high on feasibility and actionability criteria. In other words, it represents an easily implementable tool within the routinary practices of research ethics oversight bodies. As such, the proposed checklist can represent a first concrete steps towards more ambitious proposals, and could play a role not only in fostering vulnerability-sensitive reflections and practices among relevant stakeholders (including experimenters, members of ethics committees, data managers, research nurses), but can also make vulnerability a more concrete and operationalizable principle, leading to non-exclusion as well as adequate protection of research participants.

## Conclusions

The aim of this systematic review was to provide comprehensive insight into the way policy documents published internationally cover and address the concept of vulnerability in research ethics.

Despite some of the considerations pointed out in the discussion of this work had already emerged in some previous works [3 and 102], the methodological robustness of this review allowed us to obtain more accurate and reliable results that corroborate noteworthy considerations on a theme as complex as vulnerability in research ethics. Indeed, our systematic approach made it possible to analyse the concept of vulnerability from a broader perspective – one that takes into consideration all the guidelines and documents promulgated on the theme to this day.

In the past, the main documents on research ethics (such as *The Belmont Report*, the *Declaration of Helsinki*, the *GCP Guidelines* and the *CIOMS Guidelines*) stressed the importance of including vulnerable subjects in research, with the due

precautionary measures. However, this work shows that this trend has been ongoing since 1998 and that a significant number of documents include reflections on this theme. For the sake of completeness, we identified two documents that still mention the exclusion of vulnerable subjects from research, yet these should be seen as exceptions which hardly affect the larger and widely recognised trend.

Another issue established by our systematic review is that vulnerability is solidly linked to the group/population definition, rather than to a definition of the concept itself. Though perhaps pragmatically preferable, this tendency appears nonetheless problematic, and it could lead to the stigmatisation of individuals belonging to specific categories, especially when documents only include a priori definitions [101].

Some of the documents in our study pay more attention to contextual aspects. An unequal distribution of health care resources or limited access to goods and medical care are examples of contextual aspects that can exacerbate a present vulnerability.

Finally, our study sheds light on the question regarding the actual applicability (and usefulness) of the provisions contained in the various documents.

None of the seventy-nine documents in our review explicitly presents operative provisions. Although all of them, as medium-level provisions, are theoretically useful indications, they do not provide specific guidance, which proves to be problematic in concrete complex scenarios.

In spite of these issues, we trust that our results and their analysis may be considered a valuable starting point for drafting new, comprehensive and, most importantly, unambiguous policy documents, of which the "Vulnerability Checklist" here proposed may represent an – even preliminary – example. Only coherent reflections on vulnerability – its definition and its implications – will render it possible for research ethics to establish how vulnerable individuals should be treated during research.

## Strengths and limitations

The main strength of our study is its systematic methodology. We identified inclusion and exclusion criteria before proceeding in the selection of documents, in order to avoid bias. We engaged in both a pre-screening process and a screening process, following a rigorous sequence of steps (see the methodology section). These stages allowed us to obtain an extensive and inclusive list of documents from a wide range of countries worldwide. A further positive feature of this study is that it is up to date: half of the guidelines (forty-eight) were published in the last decade, with the most recent ones dating back to 2025.

In addition, the four authors of this study engaged in an ongoing mutual consultation during the analysis of all included documents, in order to discuss and resolve the various relevant details and issues. Moreover, the presence of the root "vuln-" in all documents left no room for interpretations or assumptions that might obfuscate our subject.

The main limitation of this study is perhaps that we considered only publications written in English, or officially translated in English. As a result, it is possible that there are additional policy documents written in other languages which are used to guide clinical trial practice.

## Supporting information

**S1 Table. List of documents excluded from title screening and text skimming.**
(DOCX)

**S2 Table. Documents excluded from keywords search (no keyword found, or keywords not used as a recurrent concept in the text and in a context relevant to the four research questions).**
(DOCX)

**S3 Table. Comprehensive overview integrating the most relevant findings.**
(DOCX)

**S1 Appendix. QUAGOL schemes.**
(DOCX)

**S4 Table. List of included documents with relative link.**
(DOCX)

**S5 Table. All data extracted from the primary research sources.**
(DOCX)

**S1 Checklist. Vulnerability Checklist.**
(DOCX)

## Author contributions

**Conceptualization:** Virginia Sanchini.

**Data curation:** Asia Grigis, Giorgia Beretta, Pascal Borry, Virginia Sanchini.

**Funding acquisition:** Virginia Sanchini.

**Methodology:** Virginia Sanchini.

**Project administration:** Giorgia Beretta, Virginia Sanchini.

**Supervision:** Pascal Borry, Virginia Sanchini.

**Writing – original draft:** Asia Grigis, Giorgia Beretta, Virginia Sanchini.

**Writing – review & editing:** Asia Grigis, Giorgia Beretta, Pascal Borry, Virginia Sanchini.

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
