## [Decision Letter · Decision Letter 0]

Dear Dr. Sanchini,

Comments from PLOS Editorial Office: We note that one or more reviewers has recommended that you cite specific previously published works. As always, we recommend that you please review and evaluate the requested works to determine whether they are relevant and should be cited. It is not a requirement to cite these works. We appreciate your attention to this request.

We look forward to receiving your revised manuscript.

Kind regards,

Alberto Molina Pérez, Ph.D.

Academic Editor

PLOS ONE

Journal Requirements:

2. Thank you for stating the following financial disclosure: This research is part the project “Emerging Technologies and vulnerabilities in aged care” (ElderTech), funded by Fondazione Cariplo, call “Social Research:

Science, Technology and Society”. Grant number: 2020-1322 

3. In the online submission form, you indicated that all data generated or analysed during this study are included in this published article.

Additionally, the full list of QUAGOL schema are not publicly available due to privacy

concern but are available from the corresponding author on reasonable request. 

4. As required by our policy on Data Availability, please ensure your manuscript or supplementary information includes the following: 

Reviewers' comments:

Reviewer's Responses to Questions

**Comments to the Author**

1. Is the manuscript technically sound, and do the data support the conclusions?

Reviewer #1: Yes

Reviewer #2: Yes

2. Has the statistical analysis been performed appropriately and rigorously?

Reviewer #1: Yes

Reviewer #2: N/A

3. Have the authors made all data underlying the findings in their manuscript fully available?

Reviewer #1: Yes

Reviewer #2: Yes

4. Is the manuscript presented in an intelligible fashion and written in standard English?

Reviewer #1: Yes

Reviewer #2: Yes

Reviewer #1: The systematics appears to be well conducted and the search strategy with which the search was carried out appears to be complete with relevant terminology. In my opinion, the authors should better explain the concept of disability in light of recent literature, and how it relates to that of vulnerability (for example, see doi: 10.1016/j.arrct.2020.100079; doi: 10.1007/s11196-022-09946-x).

In health care, failure to recognize a disability makes the individual even more vulnerable, increasing his or her social disadvantage. Please elaborate on this ethical aspect in the Discussion section.

Finally, the problem of the use of new digital technologies in people who are vulnerable and require state-of-the-art health care could be a detriment to the person's own dignity and human relationships with health care providers. This concept should also be mentioned by the authors among the various ethical issues raised (for example, see doi: 10.1080/17483107.2023.2288241,doi: 10.3389/fmed.2024.1437280, doi: 10.1097/MD.0000000000036671).

Reviewer #2: This manuscript presents a systematic review investigating how vulnerability is conceptualized and operationalized in research ethics policy documents. The authors analyzed 79 policy documents from guidelines/databases/grey lit, identifying recurring themes such as a tendency to define vulnerable groups rather than vulnerability itself, and a link between vulnerability and informed consent. The manuscript reveals inconsistencies in defining vulnerability (especially mostly failing to define) and identifying different vulnerable populations across different documents (resulting in unwieldy heterogeneity). There is much to like in the manuscript but, there is also room for improvement. I hope the following comments will be useful to the author(s) as they make revisions.

1. Motivation: It would be useful to motivate better why scholars and practitioners need to care about vulnerability as a regulatory category for research ethics.

a. This is especially important given that it has been critiqued as having some fundamental limitations (e.g., Levine et al 2004). In many ways, the authors’ systematic review serves to deepen concerns about the challenges with little guidance on how to improve (see my note below). They end the manuscript with a note about providing a starting point for development, but need to do more and then motivate the manuscript better. If the limitations are addressable, then the authors should preview this up front and develop in the manuscript.

b. An alternative perspective on this is that although current practice is not uniform or necessarily principled, it largely gets the job done. From this perspective, in practice vulnerability considerations act to add some redundancy to the broader principles (e.g., Belmont). So even if researchers do not address vulnerability carefully, it is not meant to be a fail-safe. In the manuscript the authors of course raise some rationale (e.g., include with protections versus exclude; etc), but could motivate the paper better.

2. Coding & Analysis

a. I really like the data coding and think that researchers need to engage seriously with more comprehensive understandings. I suspect that there is a lot that researchers could learn from the data beyond what the authors report. On that, I am actually unclear from the submission meta-information what data will be available. The authors mention privacy concerns, but it is unclear what privacy issues are in question given that these are mostly public documents? And data summarizing gated materials can be posted, even though the gated materials themselves cannot be. I encourage making as much data public as possible and more clearly discussing which data cannot be (and why).

b. It would be useful to understand better the implications of excluding non-English-language documents. The authors should consider randomly sampling a subset of papers, translating them, then coding and comparing against the English language documents.

c. The top of page 15 has an error. It says that “seventy-two documents give a definition…” but this should be seventy-two docs that identify groups/populations as vulnerable (Q2). On the bottom of page 14, it says that only ten documents gave definitions (Q1).

3. Value added and takeaways

a. There is clearly value in conducting the systematic review, especially because it is important to understand any boundary conditions of our current understanding. The current manuscript is thus valuable in helping us probe those boundaries at least to the extent that English language sources represent the broader population of documents.

b. In my view, however, the takeaways need more attention. The findings seem to echo what earlier studies have already found and discussed and the authors of this manuscript do not draw out novel implications, or if they did the novel implications currently get lost and need more attention. For examples, others have discussed:

i. categories rather than conceptualization or without justification (e.g., Luna 2009 and Findley et al 2024 who the authors do not cite);

ii. the reductionist view of vulnerability (e.g., Meek Lange et al. 2013 [Ref: #90]);

iii. the foundation versus reflective view of vulnerability (Bracken-Roche et al. 2017 [Ref: #91]);

iv. the disproportionate emphasis on consent (e.g. Hurst 2008 whom the authors do not cite)

c. Possible avenues for development:

i. I was quite intrigued by the authors note that the emphasis on categories is different from clinical ethics (page 29, including Reference #89). The authors seem to think that the difference is explained simply, which may be the case. But are there unexplored connections or opportunities for research ethics to learn from clinical ethics? It would be useful to unpack this possibility to a greater degree.

ii. The authors conclude the manuscript with a note about the need to “draft new, comprehensive, and most importantly, unambiguous policy documents” (p. 34). While it is true that some do not see vulnerability as a big concern, many do see it as such but are unclear on how to attain the ideal approach. It would be helpful if the authors could develop this point further. As they say, this is a starting point, but it remains a highly murky starting point without more concrete guidance.

1. Did any of the policy/guidance documents stand out as “model” documents that could serve as a basis for advancing the conversation?

2. Did some combination of documents provide just the right mix of factors that could serve as a basis?

3. Is there some combination of the observations the authors make in their results discussion that, if put together, provide concrete steps?

4. Are there ways in which key components of clinical ethics could map to research ethics in ways that provide more guidance?

5. Are there ways the authors could improve on what Findley et al (PNAS; 2024) proposed? After coding a large set of legislative/regulatory documents for 100+ countries, they also proposed a “starting point” framework with concrete (though preliminary) ideas and discussion of stakeholders/incentives/norms.

I fear that without developing their final point, the authors of the current manuscript will simply be read as reinforcing conventional wisdom based on a larger corpus of documents.

**Do you want your identity to be public for this peer review?** For information about this choice, including consent withdrawal, please see our Privacy Policy

Reviewer #1: No

Reviewer #2: No

---

## [Author Response · Author response to Decision Letter 1]

26 Feb 2025

5. Review Comments to the Author

Reviewer #1: The systematics appears to be well conducted and the search strategy with which the search was carried out appears to be complete with relevant terminology. In my opinion, the authors should better explain the concept of disability in light of recent literature, and how it relates to that of vulnerability (for example, see doi: 10.1016/j.arrct.2020.100079; doi: 10.1007/s11196-022-09946-x).

In health care, failure to recognize a disability makes the individual even more vulnerable, increasing his or her social disadvantage. Please elaborate on this ethical aspect in the Discussion section.

R. We thank the reviewer for this comment. We have added a few sentences and references to better specify this issue in the Discussion Section from line 603 to line 617.

Finally, the problem of the use of new digital technologies in people who are vulnerable and require state-of-the-art health care could be a detriment to the person's own dignity and human relationships with health care providers. This concept should also be mentioned by the authors among the various ethical issues raised (for example, see doi: 10.1080/17483107.2023.2288241,doi: 10.3389/fmed.2024.1437280, doi: 10.1097/MD.0000000000036671).

R. We thank the reviewer for this comment. We have added a few sentences and references to better specify this issue in the Conclusion Section from line 781 to line 797.

Reviewer #2: This manuscript presents a systematic review investigating how vulnerability is conceptualized and operationalized in research ethics policy documents. The authors analyzed 79 policy documents from guidelines/databases/grey lit, identifying recurring themes such as a tendency to define vulnerable groups rather than vulnerability itself, and a link between vulnerability and informed consent. The manuscript reveals inconsistencies in defining vulnerability (especially mostly failing to define) and identifying different vulnerable populations across different documents (resulting in unwieldy heterogeneity). There is much to like in the manuscript but, there is also room for improvement. I hope the following comments will be useful to the author(s) as they make revisions.

1. Motivation: It would be useful to motivate better why scholars and practitioners need to care about vulnerability as a regulatory category for research ethics.

a. This is especially important given that it has been critiqued as having some fundamental limitations (e.g., Levine et al 2004). In many ways, the authors’ systematic review serves to deepen concerns about the challenges with little guidance on how to improve (see my note below). They end the manuscript with a note about providing a starting point for development, but need to do more and then motivate the manuscript better. If the limitations are addressable, then the authors should preview this up front and develop in the manuscript.

b. An alternative perspective on this is that although current practice is not uniform or necessarily principled, it largely gets the job done. From this perspective, in practice vulnerability considerations act to add some redundancy to the broader principles (e.g., Belmont). So even if researchers do not address vulnerability carefully, it is not meant to be a fail-safe. In the manuscript the authors of course raise some rationale (e.g., include with protections versus exclude; etc), but could motivate the paper better.

R. We thank the reviewer for these comments, which gave us an opportunity to better motivate our work.

In our opinion the lack of clear, specific guidance on vulnerability results in ethics committees and researchers dealing with vulnerable research subjects in a sub-optimal way. Indeed, the ambiguity around vulnerability in research ethics policy documents has proven to be problematic in concrete scenarios.

Firstly, research ethics committees, in their documentation, usually require a statement about the enrolment of vulnerable populations, but they often have no expert knowledge of what this means exactly, so they rely on lists of vulnerable groups, which are the most practical guidance available. This, however, may imply that different research ethics committees use different lists, resulting in a differential treatment of research subjects, with potentially inequitable outcomes.

Secondly, researchers often have too little guidance on how to deal with vulnerable subjects, therefore they may end up excluding some populational groups defined as vulnerable, due to difficulties in managing them appropriately.

Finally, but most importantly, research subjects risk to suffer disparities or underrepresentation in research, the latter a practice that also more recent documents has strongly criticized (e.g. 2024 version of the Helsinki Declaration).

However, it is true, as the reviewer suggests, that our work clearly and unequivocally highlights the heterogeneity that exists among policy documents, and that this may deepen rather than taming existing concerns on the topic.

Despite what above, in our view this work could first serve experimenters to become more aware of the heterogeneity that exists in guidelines, so that, while regulatory actions are being carried out, they can start taking responsibility for informed, individual choices when dealing with vulnerable research subjects, in order to gradually move from a labelling approach to an analytical approach in the practice of their research.

On a higher level, this work may also serve members of research ethics committees to better navigate among different guidelines and thus refine their requests to experimenters in dealing with vulnerable research participants.

Nonetheless, we recognize that, in order to achieve this goal, more practical guidance is needed. Therefore, as the reviewer rightfully suggested, we committed to address some of the limitations of research ethics policy documents, by developing a preliminary tool for reflection on the management of vulnerable populations in research settings. We have drafted a checklist that could be used by experimenters and ethics committee members when dealing with and making decisions about vulnerable subjects (see S6 Checklist).

2. Coding & Analysis

a. I really like the data coding and think that researchers need to engage seriously with more comprehensive understandings. I suspect that there is a lot that researchers could learn from the data beyond what the authors report. On that, I am actually unclear from the submission meta-information what data will be available. The authors mention privacy concerns, but it is unclear what privacy issues are in question given that these are mostly public documents? And data summarizing gated materials can be posted, even though the gated materials themselves cannot be. I encourage making as much data public as possible and more clearly discussing which data cannot be (and why).

The full text of these policy documents is what underlies our findings. However, we also used QUAGOL schema to further extract data from our selected policy documents. The data extracted from our findings are reported in S5 Table and in S3 Appendix.

b. It would be useful to understand better the implications of excluding non-English-language documents. The authors should consider randomly sampling a subset of papers, translating them, then coding and comparing against the English language documents.

R. We thank the Reviewer for this observation; however, we believe that excluding non-English language documents did not have relevant implications on our research. Originally, the focus of this work was on international documents. On second thoughts, we decided to include also documents which were translated in English to broaden the perspective and include documents which, thanks to the translation, might have had resonance outside national borders.

However, following the reviewer observation, we reported the fact of not having included non-English language documents as one of the limitations of this study.

c. The top of page 15 has an error. It says that “seventy-two documents give a definition…” but this should be seventy-two docs that identify groups/populations as vulnerable (Q2). On the bottom of page 14, it says that only ten documents gave definitions (Q1).

R. We thank the reviewer for pointing out this mistake, which has been corrected in the revised manuscript.

3. Value added and takeaways

a. There is clearly value in conducting the systematic review, especially because it is important to understand any boundary conditions of our current understanding. The current manuscript is thus valuable in helping us probe those boundaries at least to the extent that English language sources represent the broader population of documents.

b. In my view, however, the takeaways need more attention. The findings seem to echo what earlier studies have already found and discussed and the authors of this manuscript do not draw out novel implications, or if they did the novel implications currently get lost and need more attention. For examples, others have discussed:

i. categories rather than conceptualization or without justification (e.g., Luna 2009 and Findley et al 2024 who the authors do not cite);

ii. the reductionist view of vulnerability (e.g., Meek Lange et al. 2013 [Ref: #90]);

iii. the foundation versus reflective view of vulnerability (Bracken-Roche et al. 2017 [Ref: #91]);

iv. the disproportionate emphasis on consent (e.g. Hurst 2008 whom the authors do not cite)

R. The reviewer is right. We added relevant references (Luna 2009, Hurst 2008 and Findley et al 2024) within the Discussion section.

c. Possible avenues for development:

i. I was quite intrigued by the authors note that the emphasis on categories is different from clinical ethics (page 29, including Reference #89). The authors seem to think that the difference is explained simply, which may be the case. But are there unexplored connections or opportunities for research ethics to learn from clinical ethics? It would be useful to unpack this possibility to a greater degree.

R. We thank the reviewer for these comments, and, above all, for his/her suggestions for improvements.

As to point i., the authors tacitly referred to the fact that clinical ethics (and philosophical) debates surrounding vulnerability mostly distinguish between vulnerability as a condition generally affecting human beings as such (namely, what has been labelled elsewhere as “basic human”, “intrinsic”, “ontological”, “existential”, etc., vulnerability), and vulnerability as a condition of minority and/or disadvantage affecting only some agents, due to specific situational conditions (namely, what has been labelled elsewhere as “situational”, “contingent”, “extrinsic”, etc., vulnerability). After careful consideration, we believe the reviewer is right in saying that (still unexplored) connections between conceptualisations of vulnerability in research ethics and conceptualisations of the same notion in clinical ethics may be useful, and that research ethics may be enriched by using debates and ways to frame the same notion belonging to clinical ethics. Therefore, we better clarified this connection in the Discussion section (“Vulnerability in research ethics and other domains of applied ethics”), and we also introduced the distinction between basic human (which we referred to as “fixed/inherent”) and situational (“contextual/dynamic”) vulnerability in the vulnerability checklist (see below, reply to point ii).

ii. The authors conclude the manuscript with a note about the need to “draft new, comprehensive, and most importantly, unambiguous policy documents” (p. 34). While it is true that some do not see vulnerability as a big concern, many do see it as such but are unclear on how to attain the ideal approach. It would be helpful if the authors could develop this point further. As they say, this is a starting point, but it remains a highly murky starting point without more concrete guidance.

1. Did any of the policy/guidance documents stand out as “model” documents that could serve as a basis for advancing the conversation?

2. Did some combination of documents provide just the right mix of factors that could serve as a basis?

3. Is there some combination of the observations the authors make in their results discussion that, if put together, provide concrete steps?

4. Are there ways in which key components of clinical ethics could map to research ethics in ways that provide more guidance?

5. Are there ways the authors could improve on what Findley et al (PNAS; 2024) proposed? After coding a large set of legislative/regulatory documents for 100+ countries, they also proposed a “starting point” framework with concrete (though preliminary) ideas and discussion of stakeholders/incentives/norms.

I fear that without developing their final point, the authors of the current manuscript will simply be read as reinforcing conventional wisdom based on a larger corpus of documents.

R. We really thank the reviewer for this last comment, which prompted us to improve our work not just theoretically, but also from a practical standpoint. As reported in the Result section, what this systematic review has revealed is that there is a combination of documents which inspired most policy documents and guidelines, namely: the Belmont Report, the Helsinki Declaration, CIOMS Guidelines, and GCP Guidelines. This is an important finding per se, since it may suggest that a first step for seriously considering vulnerability in research ethics would be that experimenters are somehow exposed to this set of documents and prove to know them. However, to provide more concrete guidance, we also developed a checklist (what we defined as “Vulnerability Checklist”) which, in our view, should be filled in by the experimenters and submitted to the appointed research ethics committee, along with all the documentation required for submission. In this sense, we actually recognize that the TAPIR framework proposed by Findley and colleagues represents an extraordinary contribution for scholarly debates which are interested in a broader (not only research ethics but also social and political) and non-procedural definition of vulnerability. However, insofar as it is narrower and perhaps less demanding in its operationalization, we hope that our proposed checklist will be somehow able to complement Findley et al. more comprehensive effort. (Findley et al contribution, which was not present in the original reference list as published after the first submission of this review, has now been inserted in the current version of the manuscript).

---

## [Decision Letter · Decision Letter 1]

Dear Dr. Sanchini,

Thank you for submitting your manuscript to PLOS ONE. After careful consideration, we feel that it has merit but does not fully meet PLOS ONE’s publication criteria as it currently stands. Therefore, we invite you to submit a revised version of the manuscript that addresses the points raised during the review process.

**Dear Author(s), we apologize for the prolonged review process, this is however necessitated by our rigorous review standards.  Kindly respond to the review comments from this round of reviews.  **

We look forward to receiving your revised manuscript.

Kind regards,

Patrick Ifeanyi Okonta, MBBCh, MPH, FWACS, FMCOG, MD, DRH

Academic Editor

PLOS ONE

**Journal Requirements:**

Reviewers' comments:

Reviewer's Responses to Questions

**Comments to the Author**

Reviewer #3: (No Response)

Reviewer #4: (No Response)

2. Is the manuscript technically sound, and do the data support the conclusions?

Reviewer #3: Yes

Reviewer #4: Partly

3. Has the statistical analysis been performed appropriately and rigorously?

Reviewer #3: N/A

Reviewer #4: N/A

4. Have the authors made all data underlying the findings in their manuscript fully available?

Reviewer #3: Yes

Reviewer #4: No

5. Is the manuscript presented in an intelligible fashion and written in standard English?

Reviewer #3: Yes

Reviewer #4: Yes

**Reviewer #3: ** 1. The authors have commendably addressed many of the previous concerns. However, I remain unconvinced that the revisions fully satisfy the points raised by Reviewer 2 regarding the conceptualisation and operationalisation of vulnerability. Specifically, while the authors motivated the significance of 'vulnerability' as a regulatory category in research ethics in response to the reviewer, simply noting this in the rebuttal may not be sufficient. Perhaps this crucial justification could be integrated directly into the manuscript's introduction or any other appropriate section. Without a robust and articulated rationale within the text itself, the paper's foundational argument regarding vulnerability remains potentially unpersuasive for both scholars and practitioners.

2. The proposed checklist warrants closer scrutiny. Does the data from the review inform its development, or is there evidence to support the utility of checklists in similar contexts? The authors may need to consider providing a compelling argument for how this checklist offers additional safeguards for vulnerable populations beyond the ethical considerations typically detailed in standard research protocols. As it stands, introducing a checklist risks appearing as an added burden on researchers without clearly demonstrating enhanced protection.

3. Finally, the assertion that research ethics committees often lack specific expertise regarding vulnerability, leading them to rely on lists of vulnerable groups, requires empirical support. It might be helpful to substantiate this claim with evidence from the existing literature on the functioning and expertise within research ethics review boards. Without such evidence, this statement risks undermining the credibility of current ethical review processes.

4. Line 32, 596, 599, - Use of the term/ phrase emerging themes

The current framing of theme emergence may inadvertently diminish the acknowledged role of the researcher in the analytical process. Some readers might contend that presenting themes as simply 'emerging' risks obscuring the active and interpretive work undertaken by the researcher in identifying, developing, and shaping these analytical categories. The phrasing could be perceived as downplaying the researcher's influence and agency in constructing meaning from the data.

To address this potential concern, the authors may consider explicitly articulating their role in the thematic analysis. Furthermore, incorporating a reflexivity statement would significantly strengthen the methodological rigour of the manuscript. By transparently reflecting on their potential biases, assumptions, and perspectives, the authors can provide readers with a clearer understanding of how their positionality may have shaped the data collection and interpretation process. This will enhance the trustworthiness and credibility of the findings.

5. Additional commemts

Line 632 – include page number on the reference after “to provide an analysis of vulnerability that does not render it vacuous, rescues its force and importance”

6. Line 769 -770 - The first sentence should be referenced.

7. 785-6 - Previous studies in older adults’ populations, consider “Previous studies in adult populations.”

**Reviewer #4: ** This manuscript clearly reports a systematic review that focuses on vulnerability in research ethics theory and practice. Specifically, the paper focuses on the identification and definition of vulnerable populations as contained in the selected policy documents. It is impressive that several sources of the ethics literature were employed in this review. More so, the application of the PRISMA guidelines (PRISMA-Ethics guidelines) to review the selected policy documents for a non-empirical (quantitative or qualitative) ethics literature is commendable. It is noteworthy that virtually all international guidelines and policies (in English language) that include vulnerability language were included in the review. So, actors in the drafting, implementing, and reviewing of human research protection policies regarding vulnerability across local and international research landscapes could consult the paper for reference or as a guide.

In this paper, the authors began their review by clearly stating the problem of disagreement in the meaning of vulnerability which several published literatures have equally echoed. It is important to underline the perennial challenge in arriving at a harmony in the identification, use, and application of vulnerability in research ethics. Although the paper focuses on research ethics which is clearly understandable, the restriction of the scope within the research ethics field. So, it did not cover clinical, environmental and other fields where the concept of vulnerability applies. Hopefully, readers interested in vulnerability as a general concept could expand the current understanding which is limited to research ethics. This rigorous work should be regarded as a good starting point for systematic reviews on the topic and so, could make a good resource for policy makers and those interested in vulnerability in research.

Major Issues:

The purpose of a systematic review as opposed to other forms of review or studies is to provide a summary of the existing body of and quality of the available evidence on any topic and so, the quality or the certainty of that evidence becomes a priority in the review process. So, besides the detailed review process, the quality of the review comes first in the judgment of whether the manuscript is publishable or not. Therefore, the bulk of this review centered on how the paper complied with the stated recommended format for reporting systematic reviews (of ethics and non-ethics literature).

It is of note that the authors reviewed detailed application of the subject, pointing out the different definitions and descriptions of vulnerability. This is commendable, however, beginning with the abstract, the authors should pay attention to the suggestions and where necessary provide clear rationale for not including those suggestions. The purpose of highlighting these methodological issues is to ensure that what is published afterwards can be trusted evidence which why a systematic review is recommended in the first instance.

1. The authors did not include important information in the abstract. The abstract is an important section that gives an overview of the strength of evidence of the review. Although the body of the manuscripts contain some information that is not clearly stated in the abstract, it will be helpful if the authors review each section of the abstract to see if the abstract meets either the PRISMA guidelines or the PRISMA-Ethics guidelines. https://osf.io/f3ys5/ (DOI 10.17605/OSF.IO/F3YS5).

2. Where necessary, the authors should provide line by line the reason(s) for each recommendation for reporting abstract of a systematic review of the ethics literature using PRISMA-Ethics or the original PRISMA guidelines. The following are some examples:

i. The abstract did not clearly state the objective of the systematic review. The abstract only contains, in the appropriate section, the problem the authors considered in the review but not the objective of the review. The rationale should include the problem and why it is being addressed, including the objective of the review. This information should be clearly stated in the body of the manuscript.

ii. It is challenging to judge the completeness of the literature search and how current the search was since important missing items such as inclusion and exclusion criteria, the rationale for selecting the databases, the source of the grey literature, the date of the search, the search itself (how and how many authors performed the search per stage). The authors are advised to check for the explanation and checklist of the PRISMA-Ethics/PRISMA guidelines to include missing information (in the abstract and the main section of the paper) about the search strategy.

iii. The authors should state if automation tools, Ai tools and other tools were used at any stage in the review and clearly state the tool(s), where and how they were used.

iv. Although the limitations of the study were highlighted in the main section of the manuscript, this was missing in the abstract.

v. There is no argument whether this review requires an assessment of risk of bias or grading of level of evidence; it is not likely that it requires any of these assessments but the lack of clarity as how the authors addressed sources of reporting/ selection bias and the lack of description of the steps used in each stage of the review weakens the strength of the evidence. The authors, therefore, should clearly state how they addressed different types of potential biases in this review.

vi. QUAGOL coding results are noted but the five steps employed by the authors are critical in judging the quality of the methods. To improve the quality of this review, the authors should include the description of the stages utilized for the QUAGOL method.

vii. It is not clear why grey literature was conducted only in Google Scholar while there are online sources of grey literature. It is possible that, besides using just a few databases in the search strategy, more grey literature could have been enlisted in the review where Google scholar not used as the only source of grey literature.

viii. In principle, systematic review protocols should be registered. Registration helps to maintain transparency and to avoid unnecessary study duplication by other authors. The authors might find it plausible to register the protocol used in this review. Again, increased transparency in the entire review process helps to judge the quality of evidence provided by this review.

Minor issues

1. When reviewing international policies, it is important that within the review period, there is a cross-checking of each document for updates because some inf might have changed, modified or deleted. In this review, one of the consolidated guidelines from the Office of Human Research Protections (OHRP) (see page 4, line 106-7, reference 6, page38, line.844-46), the International Compilation of Human Research Standards, has been updated. So, the 2021 edition cited in this review should be replaced with the 2024 edition. (https://www.hhs.gov/ohrp/international/compilation-human-research-standards/index.html).

2. Although this is not the focus of the study, this review on vulnerability could have been more robust if other views or notions for or against the labelling of research participants as vulnerable or in what better ways the term or concept could be used such as vulnerability as a dynamic state, a situation applicable in context and having a spectrum where there are individuals or groups at one point of the spectrum at a given time and circumstance, the under and over-inclusiveness of the term, are included in the discussion section. More so, the notion of vulnerable people as stigmatizing and those providing the reason for additional protection (such as the research ethics committees) becoming paternalistic and consequently, unfairly excluding these labeled participants, should have been highlighted. It is important to note that not only is labelling important but the situation too. More so, reviewed guidelines focus on the protection from valid informed consent may not be sufficient in protecting vulnerable participants.

Reference: Rogers WA. Vulnerability. In: Laurie G, Dove E, Ganguli-Mitra A, et al., eds. The Cambridge Handbook of Health Research Regulation. Cambridge Law Handbooks. Cambridge University Press; 2021:17-26.

**Do you want your identity to be public for this peer review?** For information about this choice, including consent withdrawal, please see our Privacy Policy

Reviewer #3: **Yes: ** Farirai Mutenherwa

Reviewer #4: **Yes: ** Chiedozie Godian Ike

---

## [Author Response · Author response to Decision Letter 2]

30 May 2025

Reviewer #3

1. The authors have commendably addressed many of the previous concerns. However, I remain unconvinced that the revisions fully satisfy the points raised by Reviewer 2 regarding the conceptualisation and operationalisation of vulnerability. Specifically, while the authors motivated the significance of 'vulnerability' as a regulatory category in research ethics in response to the reviewer, simply noting this in the rebuttal may not be sufficient. Perhaps this crucial justification could be integrated directly into the manuscript's introduction or any other appropriate section. Without a robust and articulated rationale within the text itself, the paper's foundational argument regarding vulnerability remains potentially unpersuasive for both scholars and practitioners.

R. We thank the reviewer for this comment. The reviewer was right in pointing out that the paper would have benefit from better motivating the reasons why scholars and practitioners need to care about vulnerability as a regulatory category for research ethics. We have added a few sentences and references in the Introduction section, as requested (see lines 97-109).

2. The proposed checklist warrants closer scrutiny. Does the data from the review inform its development, or is there evidence to support the utility of checklists in similar contexts? The authors may need to consider providing a compelling argument for how this checklist offers additional safeguards for vulnerable populations beyond the ethical considerations typically detailed in standard research protocols. As it stands, introducing a checklist risks appearing as an added burden on researchers without clearly demonstrating enhanced protection.

R. We thank the reviewer for this comment. In the Discussion section we explained in details to what extent the proposed checklist was informed and shaped by our research findings, therefore making explicit all the connections between study results and the checklist content (see lines 829-866).

However, although this tool was created by the authors of the review—who, through their diverse roles as researchers, study coordinators, and members of ethics committees, can in some way be considered representative of the perspectives that such a checklist should embody— the proposed checklist can only be regarded as a preliminary tool. Its actual usefulness, appropriateness, and effectiveness would require a study in which the tool is tested, something we propose to undertake in the near future should the reviewers consider the tool as potentially useful.

3. Finally, the assertion that research ethics committees often lack specific expertise regarding vulnerability, leading them to rely on lists of vulnerable groups, requires empirical support. It might be helpful to substantiate this claim with evidence from the existing literature on the functioning and expertise within research ethics review boards. Without such evidence, this statement risks undermining the credibility of current ethical review processes.

R. The reviewer is right. This statement was based on our daily experience as researchers but also as ethics committees’ members. However, since we were not able to find published evidence on this issue, we did not include in the revised manuscript this sentence, as it cannot be generalizable, and, as rightly pointed out by the reviewer, may undermine the credibility of research ethics committee’ members.

4. Line 32, 596, 599 - Use of the term/ phrase emerging themes

The current framing of theme emergence may inadvertently diminish the acknowledged role of the researcher in the analytical process. Some readers might contend that presenting themes as simply 'emerging' risks obscuring the active and interpretive work undertaken by the researcher in identifying, developing, and shaping these analytical categories. The phrasing could be perceived as downplaying the researcher's influence and agency in constructing meaning from the data.

To address this potential concern, the authors may consider explicitly articulating their role in the thematic analysis. Furthermore, incorporating a reflexivity statement would significantly strengthen the methodological rigour of the manuscript. By transparently reflecting on their potential biases, assumptions, and perspectives, the authors can provide readers with a clearer understanding of how their positionality may have shaped the data collection and interpretation process. This will enhance the trustworthiness and credibility of the findings.

R. We thank the reviewer for this very important comment. We intervened in a twofold manner. First, we made explicit that we organized the Result Section on the basis of the four research questions reported in the Materials and Method Section (see lines 275-279). Second, we briefly articulated our role in identifying themes/patterns then discussed in the Discussion section. These considerations were incorporated both in the revised abstract and in the revised main text (see Discussion section, lines 630-638).

5. Additional comments

Line 632 – include page number on the reference after “to provide an analysis of vulnerability that does not render it vacuous, rescues its force and importance”

R. Done.

6. Line 769 -770 - The first sentence should be referenced.

R. We thank the reviewer for this comment. However, after a careful assessment, we decided to delete this sentence and the related passage since no longer consistent with the reasoning flow of the revised manuscript.

785-6 - Previous studies in older adults’ populations, consider “Previous studies in adult populations.”

R. Done.

Reviewer #4

This manuscript clearly reports a systematic review that focuses on vulnerability in research ethics theory and practice. Specifically, the paper focuses on the identification and definition of vulnerable populations as contained in the selected policy documents. It is impressive that several sources of the ethics literature were employed in this review.

More so, the application of the PRISMA guidelines (PRISMA-Ethics guidelines) to review the selected policy documents for a non-empirical (quantitative or qualitative) ethics literature is commendable.

It is noteworthy that virtually all international guidelines and policies (in English language) that include vulnerability language were included in the review. So, actors in the drafting, implementing, and reviewing of human research protection policies regarding vulnerability across local and international research landscapes could consult the paper for reference or as a guide.

In this paper, the authors began their review by clearly stating the problem of disagreement in the meaning of vulnerability which several published literatures have equally echoed. It is important to underline the perennial challenge in arriving at a harmony in the identification, use, and application of vulnerability in research ethics. Although the paper focuses on research ethics which is clearly understandable, the restriction of the scope within the research ethics field. So, it did not cover clinical, environmental and other fields where the concept of vulnerability applies. Hopefully, readers interested in vulnerability as a general concept could expand the current understanding which is limited to research ethics. This rigorous work should be regarded as a good starting point for systematic reviews on the topic and so, could make a good resource for policy makers and those interested in vulnerability in research.

R. We thank the reviewer for his appreciation of our work, and we hope he will be satisfied with the revised text.

Major Issues:

The purpose of a systematic review as opposed to other forms of review or studies is to provide a summary of the existing body of and quality of the available evidence on any topic and so, the quality or the certainty of that evidence becomes a priority in the review process. So, besides the detailed review process, the quality of the review comes first in the judgment of whether the manuscript is publishable or not. Therefore, the bulk of this review centred on how the paper complied with the stated recommended format for reporting systematic reviews (of ethics and non-ethics literature).

It is of note that the authors reviewed detailed application of the subject, pointing out the different definitions and descriptions of vulnerability. This is commendable, however, beginning with the abstract, the authors should pay attention to the suggestions and where necessary provide clear rationale for not including those suggestions. The purpose of highlighting these methodological issues is to ensure that what is published afterwards can be trusted evidence which why a systematic review is recommended in the first instance.

1. The authors did not include important information in the abstract. The abstract is an important section that gives an overview of the strength of evidence of the review. Although the body of the manuscripts contain some information that is not clearly stated in the abstract, it will be helpful if the authors review each section of the abstract to see if the abstract meets either the PRISMA guidelines or the PRISMA-Ethics guidelines. https://osf.io/f3ys5/ (DOI 10.17605/OSF.IO/F3YS5).

R. The reviewer is right in pointing out that our original abstract was not enough informative. We completely revised the abstract according to PRISMA-Ethics guidelines (see the detailed reply below).

2. Where necessary, the authors should provide line by line the reason(s) for each recommendation for reporting abstract of a systematic review of the ethics literature using PRISMA-Ethics or the original PRISMA guidelines.

The following are some examples:

i. The abstract did not clearly state the objective of the systematic review. The abstract only contains, in the appropriate section, the problem the authors considered in the review but not the objective of the review. The rationale should include the problem and why it is being addressed, including the objective of the review. This information should be clearly stated in the body of the manuscript.

R. We inserted the objective of the systematic review in the revised abstract. We also included a better description of the rationale of this review in the revised body of the manuscript (see Introduction section, lines 97-109).

ii. It is challenging to judge the completeness of the literature search and how current the search was since important missing items such as inclusion and exclusion criteria, the rationale for selecting the databases, the source of the grey literature, the date of the search, the search itself (how and how many authors performed the search per stage). The authors are advised to check for the explanation and checklist of the PRISMA-Ethics/PRISMA guidelines to include missing information (in the abstract and the main section of the paper) about the search strategy.

R. We inserted the missing information in the abstract, as requested.

iii. The authors should state if automation tools, Ai tools and other tools were used at any stage in the review and clearly state the tool(s), where and how they were used.

R. We have specified in the abstract that no automation tool or other tool was used at any stage of the review.

iv. Although the limitations of the study were highlighted in the main section of the manuscript, this was missing in the abstract.

R. We have added a sentence mentioning the limitation of our study in the abstract, as rightly pointed out.

v. There is no argument whether this review requires an assessment of risk of bias or grading of level of evidence; it is not likely that it requires any of these assessments but the lack of clarity as how the authors addressed sources of reporting/ selection bias and the lack of description of the steps used in each stage of the review weakens the strength of the evidence. The authors, therefore, should clearly state how they addressed different types of potential biases in this review.

R. To address and minimise potential sources of bias in the selection and analysis of documents, we set up several strategies.

First, the authors a priori defined inclusion and exclusion criteria, as reported in the “Materials and Method” Section.

Secondly, authors independently screened the included documents. Document selection was conducted by the two co-first authors (AG and GB), in consultation with the last author (VS).

Data extraction was performed using the standardised QUAGOL methodology, applied on the whole set of documents.

Given the specific type of the sources (policy documents and published guidelines), in our view, no formal risk of bias tool (e.g., AGREE II, AACODS) was applicable.

However, the credibility of the sources was assessed based on issuing authority, publication date, scope, and relevance to our research questions.

Limitations of this approach are reported in the “Strength and limitations” Section.

vi. QUAGOL coding results are noted but the five steps employed by the authors are critical in judging the quality of the methods. To improve the quality of this review, the authors should include the description of the stages utilized for the QUAGOL method.

R. We thank the reviewer for this comment. We have added a brief description of the stages of the QUAGOL methodology followed in this review in the “Materials and Method” Section (see lines 221-228) as well as in the revised abstract.

vii. It is not clear why grey literature was conducted only in Google Scholar while there are online sources of grey literature. It is possible that, besides using just a few databases in the search strategy, more grey literature could have been enlisted in the review where Google scholar not used as the only source of grey literature.

R. We consulted a grey literature database exclusively to conduct an additional verification, aimed at ensuring the inclusion of all relevant international documents either originally published in English or translated into English on the topic. While the three compiled lists, when considered jointly, should be comprehensive, we nonetheless opted to strengthen the rigor of our approach by supplementing them with a targeted search both in a bibliographic database and in grey literature, using Google Scholar, which is our standard tool for such tasks. Should the reviewer consider it necessary, we remain available to conduct an additional search in a different grey literature database, although we are confident that no further relevant documents would be identified.

viii. In principle, systematic review protocols should be registered. Registration helps to maintain transparency and to avoid unnecessary study duplication by other authors. The authors might find it plausible to register the protocol used in this review. Again, increased transparency in the entire review process helps to judge the quality of evidence provided by this review.

R. The reviewer is right in highlighting the importance of registering systematic reviews in public databases, as this contributes to greater transparency. We attempted to register our systematic review in PROSPERO as suggested; however, during the submission process, we realized that it was no longer possible to proceed due to time constraints. As stated in point 5 of the PROSPERO guidelines: "Please note: Reviews that have progressed beyond the point of completing data extraction at the time of initial registration are not eligible for inclusion in PROSPERO." We noticed that this applies to other databases as well. We apologize for this and will take it into consideration for future systematic review projects.

Minor issues

1. When reviewing international policies, it is important that within the review period, there is a cross-checking of each document for updates because some inf might have changed, modified or deleted. In this review, one of the consolidated guidelines from the Office of Human Research Protections (OHRP) (see page 4, line 106-7, reference 6, page38, line.844-46), the International Compilation of Human Research Standards, has been updated. So, the 2021 edition cited in this review should be replaced with the 2024 edi

---

## [Editor Report · Decision Letter 2]

Vulnerability in research ethics. A systematic review of policy guidelines and documents.

PONE-D-24-28751R2

Dear Dr. Sanchini,

We’re pleased to inform you that your manuscript has been judged scientifically suitable for publication and will be formally accepted for publication once it meets all outstanding technical requirements.

Kind regards,

Patrick Ifeanyi Okonta, MBBCh, MPH, FWACS, FMCOG, MD, DRH

Academic Editor

PLOS ONE
---

## [Editor Report · Acceptance letter]

PONE-D-24-28751R2

PLOS ONE

Dear Dr. Sanchini,

I'm pleased to inform you that your manuscript has been deemed suitable for publication in PLOS ONE. Congratulations! Your manuscript is now being handed over to our production team.

Kind regards,

on behalf of

Professor Patrick Ifeanyi Okonta

Academic Editor

PLOS ONE